# Regime shift detection and neurocomputational substrates for under and overreactions to change

**Mu-Chen Wang[1]\*, George Wu[2], Shih-Wei Wu[1,3]\***

[1]Institute of Neuroscience, National Yang Ming Chiao Tung University, Taipei, Taiwan; [2]Booth School of Business, University of Chicago, Chicago, United States; [3]Brain Research Center, National Yang Ming Chiao Tung University, Taipei, Taiwan

## eLife Assessment

This study offers **valuable** insights into how humans detect and adapt to regime shifts, highlighting dissociable contributions of the frontoparietal network and ventromedial prefrontal cortex to sensitivity to signal diagnosticity and transition probabilities. The combination of an innovative instructed-probability task, Bayesian behavioral modeling, and model-based fMRI analyses provides **solid** support for the main claims. The addition of new model-comparison figures in revision effectively addresses the previously noted potential confound between posterior switch probability and time in the neuroimaging results. At the behavioral level, while the computational model captures the pattern of "system neglect" well, qualitatively distinct mechanisms, such as hyper-prior attraction toward experiment-wise mean parameters, reporting biases, or probability-outlier underweighting, could produce similar behavioral signatures and cannot be fully disambiguated with the current design alone; however, converging evidence from the authors' prior work partially mitigates this concern.

**\*For correspondence:**
raccoon65.y@nycu.edu.tw (M-CW);
swwu@nycu.edu.tw (S-WW)

**Competing interest:** The authors declare that no competing interests exist.

**Abstract** The world constantly changes, with the underlying state of the world shifting from one regime to another. The ability to detect a regime shift, such as the onset of a pandemic or the end of a recession, significantly impacts individual decisions, as well as governmental policies. However, determining whether a regime has changed is usually not obvious, as signals are noisy and reflective of the volatility of the environment. We designed an fMRI paradigm that examines a stylized regime-shift detection task. Human participants showed systematic overreaction and underreaction: Overreaction was most commonly seen when signals were noisy, but when environments were stable and change is possible but unlikely. By contrast, underreaction was observed when signals were precise but when environments were unstable and hence change was more likely. These behavioral signatures are consistent with the *system-neglect* computational hypothesis, which posits that sensitivity or lack thereof to system parameters (noise and volatility) is central to these behavioral biases. Guided by this computational framework, we found that individual subjects' sensitivity to system parameters was represented by two distinct brain networks. Whereas a frontoparietal network selectively represented individuals' sensitivity to signal noise but not environment volatility, the ventromedial prefrontal cortex (vmPFC) showed the opposite pattern. Further, these two networks were involved in different aspects of regime-shift computations: while vmPFC correlated with subjects' beliefs about change, the frontoparietal network represented the strength of evidence in favor of regime shifts. Together, these results suggest that regime-shift detection recruits belief-updating and evidence-evaluation networks and that under- and overreactions arise from how sensitive these networks are to the system parameters.

## Introduction

Judging whether the world has changed is ubiquitous, from public health officials grappling with whether a pandemic surge has peaked, central banks figuring out whether inflation is easing, investors discerning whether the electric car market is getting traction, or romantic partners divining whether a relationship has soured. In all of these examples, individuals must update their beliefs that the world has changed based on a noisy signal, such as a drop in positive pandemic cases or a romantic partner's suddenly mysterious behavior. In some cases, epidemiological or statistical models provide guidance. However, in many, if not most cases, the determination of whether a regime shift has occurred is made intuitively (*Sanders and Manrodt, 2003*).

We investigate intuitive judgments of regime-shift detection using a simple empirical paradigm (*Massey and Wu, 2005*; *Seifert et al., 2023*). Although this paradigm abstracts away some complications of real-world change detection, it maintains the most central features of the problem: normatively, regime-shift judgments reflect the *signals* from the environment as well as knowledge about the *system* that produces the signals. The most recent time series of inflation rates, pandemic cases, and sales of electric cars are all examples of signals. When pandemic cases continue to decline in recent weeks, one might infer a shift from pandemic to non-pandemic regime, only to learn a few weeks later that pandemic has resurged. Indeed, signals such as the latest pandemic cases are seldom precise indications of the true state of the world. Put differently, signals are, by and large, noisy. The noisier the signals are, the less *diagnostic* they are of the underlying regime.

In addition, signals are affected by how likely the regime shifts from one to another (transition probability). These two fundamental features or *system parameters*—the diagnosticity of the signals and transition probability—can be conceptualized as two independent aspects of the system that generates the signals. Previous works on regime-shift detection have found that people tend to overreact to change when they receive noisy signals (low signal diagnosticity) but nonetheless are in a stable environment (small transition probability). By contrast, precise signals (high signal diagnosticity) in an unstable environment (large transition probability) typically result in underreaction (*Benjamin, 2019*; *Brown and Steyvers, 2009*; *Massey and Wu, 2005*).

*Massey and Wu, 2005* proposed that over- and underreactions reflect *system neglect*—the tendency to respond primarily to signals and secondarily to the system parameters that produce the signals. The system-neglect hypothesis was derived from theoretical accounts of the determinants of confidence by *Griffin and Tversky, 1992*. To explain system neglect, consider someone who is making judgments on whether a stock market had shifted from the bear to the bull market regime and has been given information about recent stock returns (signals), how frequent regime shifts happen (transition probability), and how similar the two regimes are (signal diagnosticity). If her judgments are solely based on the signals and not affected by transition probability and signal diagnosticity, she shows a *complete* neglect of the system parameters. Broadly, system neglect describes a lack of sensitivity—compared with normative Bayesian updating—to the system parameters. In the case of regime-shift detection, this leads to insufficient belief revision (i.e. underreaction) in diagnostic and unstable environments, where Bayesian updating requires a larger change in beliefs, and excessive belief change in noisy and stable environments (i.e. overreaction), where Bayesian updating calls for less pronounced belief revision. Empirical patterns akin to system neglect are not only observed in regime-shift detection, but also in other domains such as confidence judgments (*Griffin and Tversky, 1992*; *Kraemer and Weber, 2004*), demand forecasting (*Kremer et al., 2011*), and pricing decisions (*Seifert et al., 2023*). Under- and overreactions have been an active research topic in financial economics, often measured as reactions to stock market changes or firm news (*Baker and Wurgler, 2007*; *Barberis et al., 1998*; *Daniel et al., 1998*; *De Bondt and Thaler, 1985*; *Nelson et al., 2001*).

At the neurobiological level, change detection has been investigated in the context of reinforcement learning in dynamic environments where changes in the state of the world, such as reward distributions, take place during the experiments (*Soltani and Izquierdo, 2019*). Different behavioral paradigms, most notably reversal learning, and computational models were developed to investigate its neurocomputational substrates (*Behrens et al., 2007*; *Izquierdo et al., 2017*; *McGuire et al., 2014*; *Muller et al., 2019*; *Nassar et al., 2010*; *Payzan-LeNestour and Bossaerts, 2011*; *Payzan-LeNestour et al., 2013*). Key findings on the neural implementations for such learning include identifying brain areas and networks that track volatility in the environment (rate of change; *Behrens et al., 2007*), the uncertainty or entropy of the current state of the environment (*Muller et al., 2019*),

participants' beliefs about change (*Kao et al., 2020*; *McGuire et al., 2014*; *Payzan-LeNestour and Bossaerts, 2011*), and their uncertainty about whether a change had occurred (*Kao et al., 2020*; *McGuire et al., 2014*). Evidence from several of the aforementioned studies (*Behrens et al., 2007*; *Kao et al., 2020*; *McGuire et al., 2014*) suggests that the dorsomedial frontal cortex (DMFC) is critical to learning in dynamic environments, as information about volatility, subjective beliefs, and uncertainty about change converge in this brain region.

But how do biases in change detection arise in the brain? Although reinforcement learning studies provide valuable insights into change detection in the learning process, it remains unclear how biases in change detection—under- and overreactions to change—arise at the neural algorithmic and implementation levels. For example, it is unclear how a certain brain area, such as DMFC, that had been shown to represent environmental volatility, would contribute to under- and overreactions to change. In order to systematically characterize under- and overreactions, it would be critical to (1) adopt a well-established behavioral paradigm that robustly elicits these behavioral phenomena and (2) have computational frameworks suitable for developing neural hypotheses regarding under- and overreactions. To address these issues, in this study, we adopted the regime-shift detection task from *Massey and Wu, 2005* and their system-neglect computational framework. At the behavioral level, the regime-shift task is a well-established paradigm that robustly elicits under- and overreactions to change. At the algorithmic and implementation levels, the system-neglect framework provides a straightforward neurocomputational hypothesis regarding under- and overreactions. It predicts that for brain areas involved in regime-shift detection, under- and overreactions arise from their sensitivity or lack thereof in response to the system parameters.

We replicated previous behavioral findings on under- and overreactions (*Massey and Wu, 2005*). Using blood-oxygen-level-dependent (BOLD) functional magnetic resonance imaging (fMRI), we reported three key findings. First, we identified two distinct brain networks involved in regime-shift detection, with the ventromedial prefrontal cortex (vmPFC) and ventral striatum in representing subjects' reported beliefs about change and a frontoparietal network in evaluating the strength of change evidence. Second, we found that these two networks selectively respond to different system parameters: while the frontoparietal network represents individual subjects' sensitivity to signal diagnosticity but not transition probability, the vmPFC shows the opposite pattern. Third, the neural sensitivity profiles were signal-dependent: the frontoparietal network only represented individuals' sensitivity to signal diagnosticity when signals consistent with change appeared. By contrast, vmPFC represented individuals' sensitivity to transition probability regardless of whether subjects received signals consistent or inconsistent with change. Such signal-dependent representations led us to further examine and subsequently verify that they are indeed key properties of our system-neglect computational model. Together, these results suggest that regime-shift detection is implemented jointly by a belief-updating network (vmPFC-striatum) and evidence evaluation network (frontoparietal network) and that their sensitivity in response to different environmental parameters contributes to under- and overreactions to change. More broadly, we showed that neural data can reveal important properties of computational models that are overlooked in theoretical treatments and behavioral analyses.

## Results

In our *regime-shift detection* task (*Figure 1A*), in each trial, subjects saw a series of sequentially presented sensory signals (red or blue balls). They were told that the signals came from one of two regimes, the red regime or the blue regime (*Figure 1B*). Regimes were symmetric, for example, with a red regime consisting of 60 red balls and 40 blue balls and the corresponding blue regime consisting of 60 blue balls and 40 red balls. Each trial started with the red regime but could shift to the blue regime before each of the 10 periods in a trial. After seeing a new signal in each period, subjects provided a probability estimate that the current regime was the blue regime, that is a posterior probability of a regime shift. They were also instructed that once the regime has shifted from the red to the blue during a trial, the regime would remain in the blue regime until the end of the trial, that is the blue regime was a trapping or absorbing state. Our experimental paradigm hence follows *Massey and Wu, 2005*. Note that, during a trial, subjects did not receive feedback—after making probability estimates in each period—on whether the regime had shifted and the monetary bonus earned as a result of accuracy in probability estimates (see *Materials and methods* for details). Hence, subjects had no access to information about accuracy and rewards as she or he was making probability estimates.

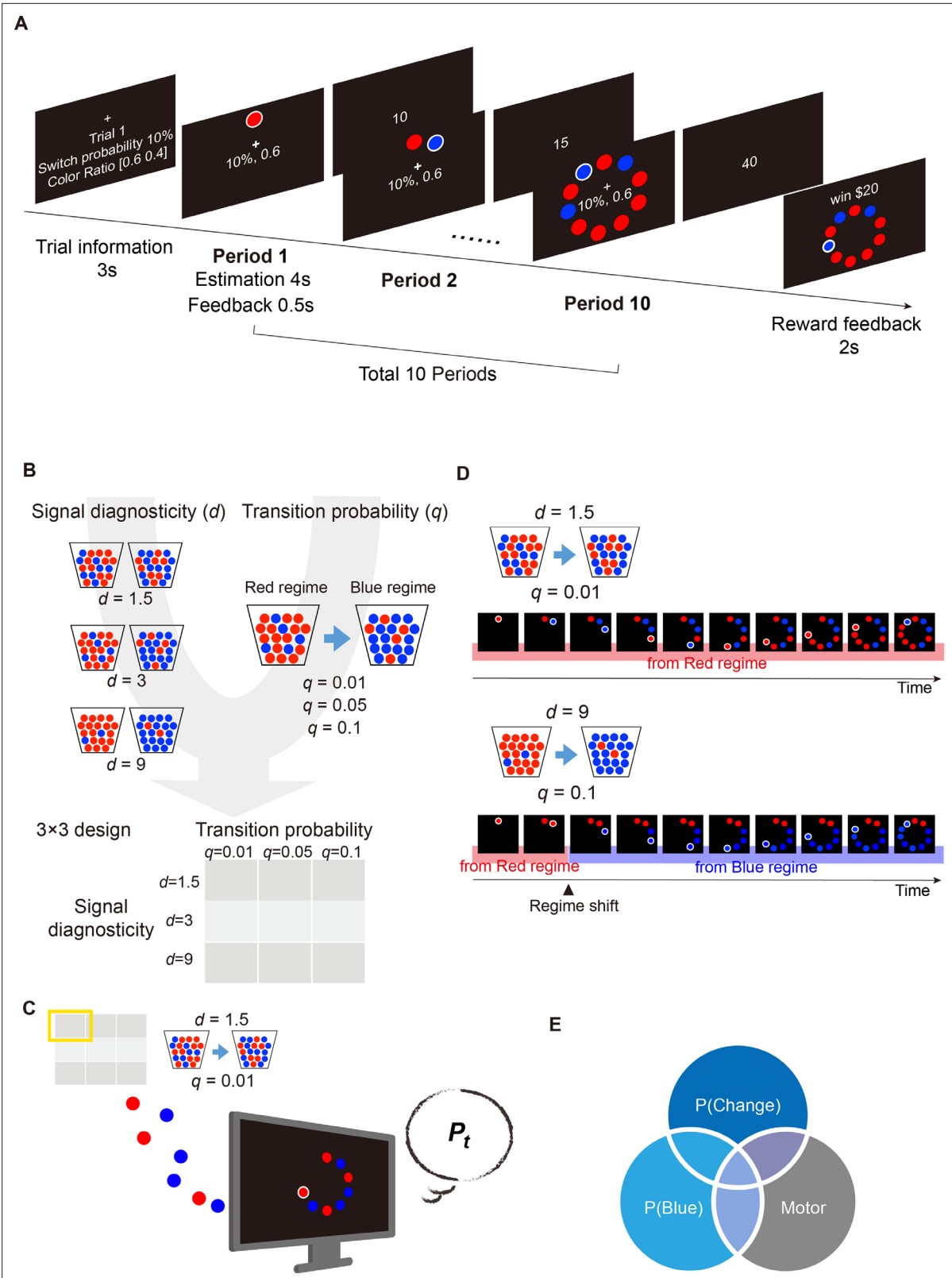

**Figure 1.** The regime-shift detection task. (**A**) Trial sequence. In each trial, the subjects saw a sequence of red and/or blue signals and were told that these signals were drawn from one of the two regimes, a Red regime and a Blue regime. Both regimes were described as urns containing red and blue balls. The Red regime contained more red balls, while the Blue regime contained more blue balls. Each trial always started at the Red regime but could shift to the Blue regime in any of the 10 periods according to some transition probability (q). At the beginning of a trial, information about transition

*Figure 1 continued on next page*

*Figure 1 continued*

probability (shown as 'switch' probability in the illustration) and signal diagnosticity (shown as 'color ratio') were revealed to the subjects. In this example, the transition probability is 0.1 and signal diagnosticity is 1.5. See main text for more detailed descriptions. (**B**) Manipulation of the system parameters, that is transition probability (**q**) and signal diagnosticity (**d**). We independently manipulated the $q$ (3 levels) and $d$ (3 levels), resulting in a 3×3 factorial design. (**C**) An example of a particular combination of the system parameters from the 3×3 design. Here, the system that produces the signals has a $q=0.01$ transition probability and $d=1.5$ signal diagnosticity. Signals were sequentially presented to subjects. After each new signal appeared (a period), subjects provided a probability estimate ($P_t$) of a regime shift. (**D**) Two example trials sequences. The example on the left shows the sequence of 10 periods of blue and red signals where $d=1.5$ and $q=0.01$. In this example, the regime was never shifted. The example on the right shows the sequence of periods where $d=9$ and $q=.1$. In this example, the regime was shifted from the Red to the Blue regime in Period 3 such that the signals shown starting at this period were drawn from the Blue regime. (**E**) We performed three fMRI experiments (30 subjects in each experiment) to investigate the neural basis of regime-shift judgments. Experiment 1 was the main experiment looking at regime shift—which corresponds to $P\left(Change\right)$ in the Venn diagram—while Experiments 2 and 3 were the control experiments that ruled out additional confounds. In both Experiments 1 and 2, the subjects had to estimate the probability that signals came from the blue regime. But unlike Experiment 1, in Experiment 2, which corresponds to $P\left(Blue\right)$, no regime shift was possible. In Experiment 3, the subjects were simply asked to enter a number with a button-press setup identical to Experiments 1 and 2. Therefore, Experiment 3 (Motor) allowed us to rule out motor confounds.

We manipulated two system parameters, transition probability and signal diagnosticity (*Figure 1BC*). Transition probability, $q$, with possible values of 0.01, 0.05, and 0.1, specified the probability that the regime would shift from the red to the blue regime in any period. Signal diagnosticity, $d$, with possible values of 1.5, 3, and 9, captured the degree to which the two regimes differed. For example, an environment with high signal diagnosticity (e.g. $d=9$) indicated that there were nine times more red balls than blue balls in the red regime (a 90:10 Red to Blue ratio) and nine times more blue balls than the red balls in the blue regime (a 90:10 Blue to Red ratio). Therefore, the weight that a signal (blue or red ball) carried was captured by the signal diagnosticity: in a low diagnostic environment, a blue signal most likely reflects no change in regime ($d=1.5$, example on the top of *Figure 1D*). By contrast, in a highly diagnostic environment, a blue signal very likely reveals a shift in regime ($d=9$, example on the bottom of *Figure 1D*). At the beginning of each trial, subjects were informed about the transition probability and signal diagnosticity in that trial. In the example trial sequence (*Figure 1A*), the transition probability (indicated by 'switch probability' in *Figure 1A*) is 0.1 while the signal diagnosticity (indicated by 'color ratio' in *Figure 1A*) is 1.5, with the red regime consisting of 60 red balls and 40 blue balls and the blue regime consisting of 40 red balls and 60 blue balls.

To establish the neural representations for regime-shift estimation, we performed three fMRI experiments ($n=30$ subjects for each experiment, 90 subjects in total). Experiment 1 was the main experiment, while Experiments 2–3 were control experiments that ruled out two important confounds (*Figure 1E*). The control experiments were designed to clarify whether any effect of subjects' probability estimates of a regime shift, $P_t$, in brain activity can be uniquely attributed to change detection. Here we considered two major confounds that can contribute to the effect of $P_t$. First, since subjects in Experiment 1 made judgments about the probability that the current regime is the blue regime (which corresponded to the probability of regime change), the effect of $P_t$ did not particularly have to do with change detection. To address this issue, in Experiment 2 subjects made exactly the same judgments as in Experiment 1 except that the environments were stationary (no transition from one regime to another was possible), as in *Edwards, 1968* classic 'bookbag-and-poker chip' studies. Subjects in both experiments had to estimate the probability that the current regime is the blue regime, but this estimation corresponded to the estimates of regime change only in Experiment 1. Therefore, activity that correlated with probability estimates in Experiment 1 but not in Experiment 2 can be uniquely attributed to representing regime-shift judgments. Second, the effect of $P_t$ can be due to motor preparation and/or execution, as subjects in Experiment 1 entered two-digit numbers with button presses to indicate their probability estimates. To address this issue, in Experiment 3, subjects performed a task where they were presented with two-digit numbers and were instructed to enter the numbers with button presses. By comparing the fMRI results of these experiments, we were therefore able to establish the neural representations that can be uniquely attributed to the probability estimates of regime-shift.

## Behavioral evidence for over- and underreactions to change

Our analyses used subjects' probability estimates of a regime shift, $P_t$, for each period, $t=1,\ldots,10$. We found that subjects were in general responsive to the system parameters, with higher $P_t$ when

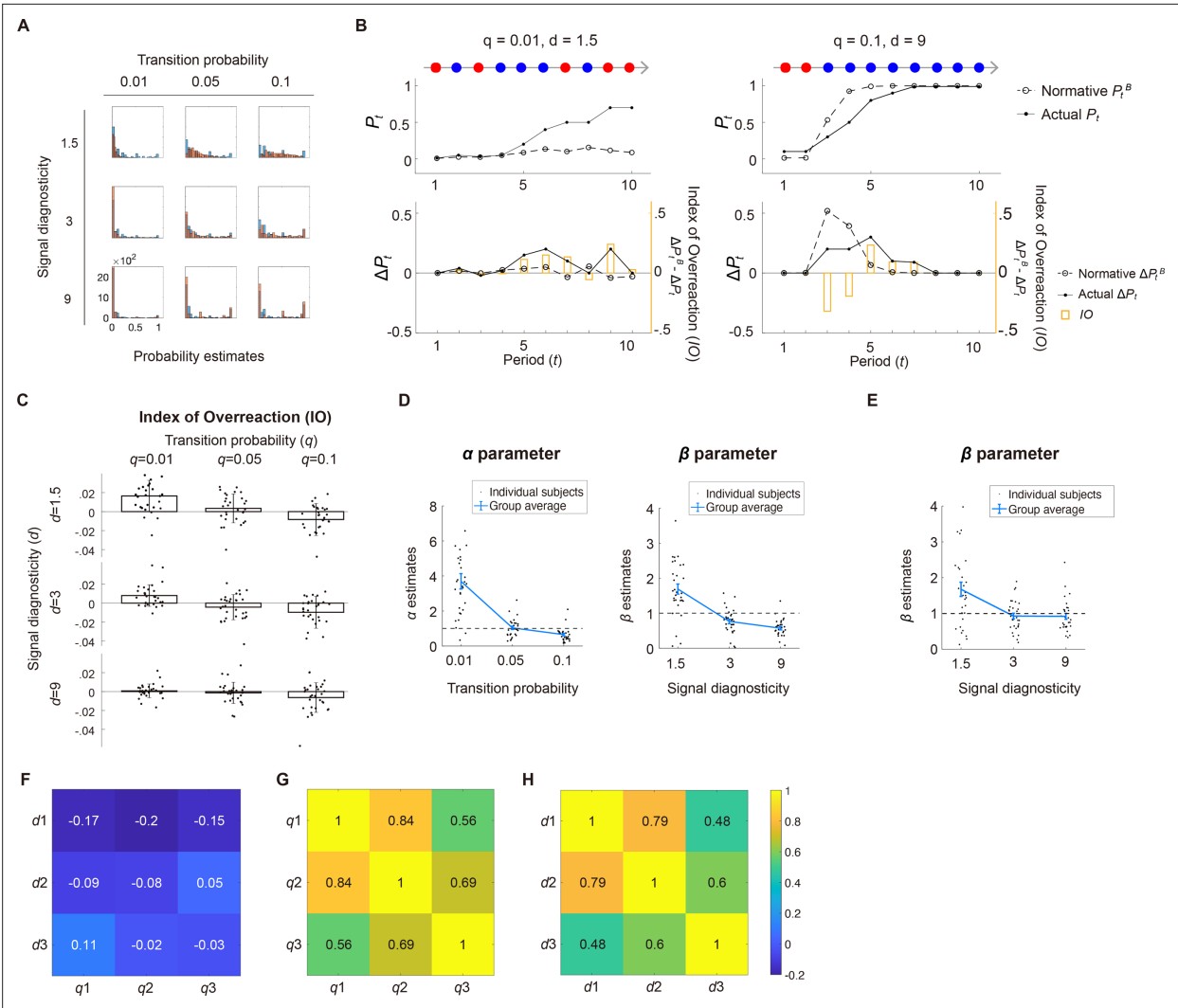

**Figure 2.** Behavioral results. (**A**) Probability estimates ($P_t$) from all subjects are plotted as histograms separately for each condition—a combination of transition probability and signal diagnosticity. The blue bars represent the actual probability estimates, while the orange bars correspond to the probability estimates predicted by the Bayesian model. (**B**) Illustrations of over- and underreactions. Left column: stable environment ($q=0.01$) with noisy signals ($d=1.5$) and the 10 periods of red and blue signals a subject encountered. Right column: unstable environment ($q=0.1$) with precise signals ($d=9$). Top row: we plot a subject's actual probability estimates ($P_t$, solid line) and the normative Bayesian posterior probability ($P_t^B$, dashed line). Bottom row: belief revision shown by the subject ($\Delta P_t = P_t, -P_{t-1}$, solid line) and the Bayesian belief revision ($\Delta P_t^B$, dashed line). The orange bars represent $\Delta P_t - \Delta P_t^B$, which we define as the Index of Overreaction ($IO$; vertical axis in orange on the right). (**C**) Over- and underreactions to change (Experiment 1). The mean $IO$ (across all 30 subjects) is plotted as a function of transition probability and signal diagnosticity. Subjects overreacted to change if $IO>0$ and underreacted if $IO<0$. (**D**) Parameter estimates of the system-neglect model (Experiment 1). Left graph: Weighting parameter ($\alpha$) for transition probability. Right graph: Weighting parameter ($\beta$) for signal diagnosticity. Dashed lines indicate parameter values equal to 1, which is required for Bayesian updating. (**E**) Parameter estimates of the system-neglect model (Experiment 2). Weighting parameter ($\beta$) for signal diagnosticity in the system-neglect model. (**F**) Correlation between $\alpha$ and $\beta$ estimates at different levels of transition probability ($q_1$ to $q_3$) and signal diagnosticity ($d_1$ to $d_3$). All pairwise Pearson correlation coefficients (indicated by the values on the table that were also color coded) were not significantly different from 0 ($p > 0.05$). (**G**) Pearson correlation coefficients of $\alpha$ estimates between different levels of transition probability. All pairwise correlations were significantly different from 0 ($p < 0.05$). (**H**) Pearson correlation coefficients of $\beta$ estimates between different levels of signal diagnosticity. All pairwise correlations were significantly different from 0 ($p < 0.05$). Error bars represent ±1 standard error of the mean (n=30).

the transition probability was larger (**Figure 2A**). We also found that subjects tended to give more extreme $P_t$ under high signal diagnosticity than low diagnosticity (**Figure 2A**). In addition, we used a measure of belief revision, $\Delta P_t=P_t - P_{t-1}$. In **Figure 2B**, we show examples of $P_t$ and $\Delta P_t$ from a subject. On the left, the subject was in a stable environment (small transition probability, $q=0.01$) and faced two regimes that were very similar to each other (low signal diagnosticity, $d=1.5$). The red and

blue signals (10 periods) were what the subject encountered during a trial. On the right, the subject was in an unstable environment ($q=0.1$) and faced two regimes that were very different ($d=9$).

To examine over- and underreactions to change, we compared subjects' belief revision, $\Delta P_t = P_t - P_{t-1}, t=2, \ldots, 10$, with belief revision predicted by the Bayesian model, $\Delta P_t^B = P_t^B - P_{t-1}^B$ (see *Figure 2B* for illustrations). $\Delta P_t$ and $\Delta P_t^B$, respectively, capture how much subjects and a normative Bayesian change probability estimates in response to a new signal. When $\Delta P_t > \Delta P_t^B$, it indicates larger belief revision than the normative Bayesian, that is an overreaction. By contrast, $\Delta P_t < \Delta P_t^B$ indicates smaller belief revision, that is an underreaction. We therefore use $IO = \Delta P_t - \Delta P_t^B$ as an Index of Over-reaction ($IO$). We found that subjects tended to overreact to change ($IO > 0$) when they received noisy signals (i.e. low signal diagnosticity, $d=1.5$) and when the environment was stable (small transition probability, $q=0.01$). By contrast, underreaction ($IO < 0$) was most commonly observed when they were in unstable environments (large transition probability, $q=0.1$) and with clear signals (i.e. high signal diagnosticity, $d=9$; *Figure 2C*). These patterns of over- and underreactions were consistent with findings in *Massey and Wu, 2005* and the system-neglect hypothesis, which posits a tendency to respond primarily to the signals and secondarily to the system that generates the signals (*Massey and Wu, 2005*; *Seifert et al., 2023*). According to the system-neglect hypothesis, responding secondarily to the system is synonymous with a lack of sensitivity to the system parameters, which leads to underreactions in unstable environments with precise signals and overreactions in stable environments with noisy signals.

Following *Massey and Wu, 2005*, we quantitatively model these belief revisions using the system-neglect model (see *System-neglect model* in *Methods*). The model is a parameterized version of the normative Bayesian model that allows for distortion of the system parameters via weighting parameters for transition probability ($\alpha$) and signal diagnosticity ($\beta$). In short, $\alpha$ reflects distortion of transition probability, with $\alpha \times q$ in the system-neglect model capturing a decision maker's *effective* transition probability ($q$). For example, if $\alpha=4$ when $q=0.01$, the decision maker effectively treats a 0.01 transition probability as if it were 0.04. By contrast, $\beta$ captures the extent to which the decision maker overweighs or underweighs signal diagnosticity ($d^\beta$) when faced with a signal. For example, if $\beta=2$ when $d=1.5$, subjects would treat a blue signal by updating the odds ratio for change by $1.5^2$, or 2.25 rather than 1.5.

In the system-neglect model, we estimated the weighting parameters separately for each level of transition probability and signal diagnosticity, that is $\alpha_i \times q_i$ and $d_j^{\beta_j}$, where $\alpha_1$, $\alpha_2$, and $\alpha_3$ correspond to transition probabilities of 0.01, 0.05, and 0.1, respectively, and $\beta_1$, $\beta_2$, and $\beta_3$ correspond to signal diagnosticity of 1.5, 3, and 9, respectively. In contrast to the Bayesian model which implies $\alpha_i = \beta_j = 1$ for all $i, j$, the system-neglect model requires that $\alpha_i > \alpha_{i+1}$ and $\beta_j > \beta_{j+1}$ because it would effectively capture a lack of sensitivity to the system parameters.

We fit the model to $P_t$ for each subject separately and found parameter estimates consistent with system neglect (*Figure 2D*). The mean estimates of $\alpha$ were 3.69, 1.04, and 0.65 for $q=0.01$, 0.05 and 0.10, respectively. The parameters indicated that, on average, when $q=0.01$, subjects treated as if it were 0.0369. By contrast, when $q=0.10$, the subjects treated it as if it were 0.065. Thus, a factor of 10 in actual transition probability (0.01 vs 0.1) was reduced to a factor of less than 2 (0.0369 vs 0.065) in effective transition probability. For signal diagnosticity, the mean parameter estimates of $\beta$ were 1.69, 0.77, and 0.57 for $d=1.5$, 3, and 9, respectively. Thus, subjects updated their beliefs $1.5^{1.69}=1.98$ when $d_1=1.5$ and $9^{0.57}=3.50$ when $d_3=9$. Normatively, the change in odds ratio between the two conditions should have been $d_3/d_1=6$ but, consistent with system neglect, was considerably smaller, $3.50/1.98=1.76$. Together, large parameter estimates ($\alpha > 1, \beta > 1$) at low signal diagnosticity (noisy signals) and low transition probability (stable environments) capture overreactions to changes, while small parameter estimates ($\alpha < 1, \beta < 1$) at large signal diagnosticity (precise signals) and large transition probability (unstable environments) reflect underreactions to change. These results replicate the findings by *Massey and Wu, 2005*, with the pattern of over- and underreactions as predicted by the system neglect hypothesis. Critically, the degree of system neglect can be captured by the negative trend of the parameter estimate as a function of the system parameter levels (*Figure 2D*): the steeper the slope, the larger the system neglect. We found a similar pattern on $\beta$ in Experiment 2 (one of the control experiments) where environments were stationary (no transition probability) and signal diagnosticity was manipulated (*Figure 2E*; *Benjamin, 2019*; *Tversky et al., 1990*).

We next examined whether the way subjects respond to different system parameters is similar. It is possible, for example, that subjects who showed stronger (or weaker) distortion of transition probability (captured by $\alpha$ parameter) also showed stronger (or weaker) distortion of signal diagnosticity (captured by $\beta$ parameter). There was no significant correlation between $\alpha$ and $\beta$ parameters (*Figure 2F*). However, we did find within-parameter correlation: subjects who had a higher $\alpha_i$ for a transition probability level $i$ also tended to have a higher $\alpha_{i'}$ for a second transition probability level $i'$ (*Figure 2G*), with the same pattern also holding for signal diagnosticity parameter $\beta$ (*Figure 2H*). Together, these results suggested that the way an individual decision maker responds to information about the probability of change in the environment (transition probability) has little to do with how she or he responds to information about the similarity between different regimes (signal diagnosticity). But individuals are consistent in responding to a particular system parameter (transition probability or signal diagnosticity) across different levels of the parameter.

We performed parameter recovery analysis to examine whether the fitting procedure gave reasonable parameter estimates (*Wilson and Collins, 2019*). First, we simulated each subject's probability estimation data based on the system-neglect model by using that subject's parameter estimates.

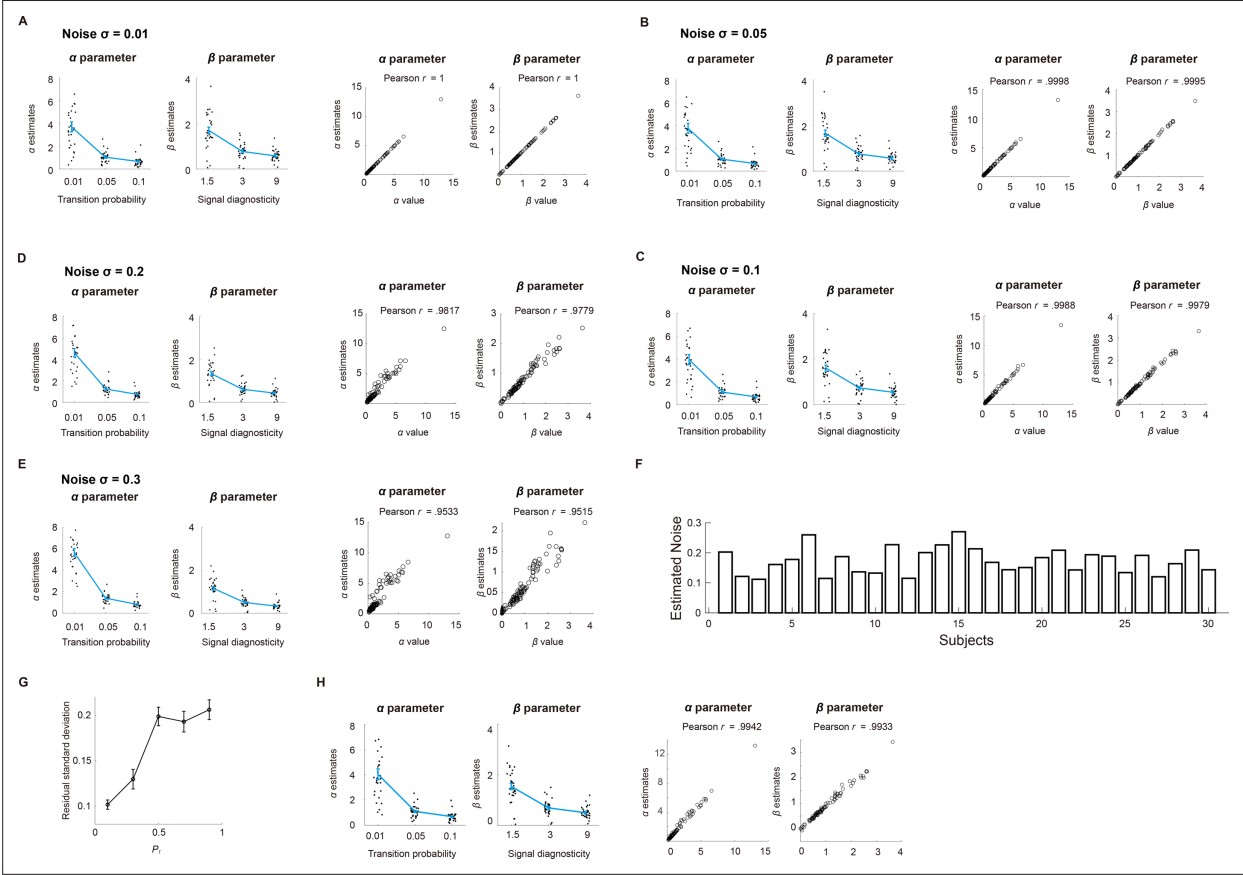

**Figure 3.** Parameter recovery analysis. We simulated probability estimates according to the system-neglect model. We used each subject's parameter estimates as our choice of parameter values used in the simulation. Using simulated data, we estimated the parameters ($\alpha$ and $\beta$) in the system-neglect model. To examine parameter recovery, we plot the parameter values we used to simulate the data against the parameter estimates we obtained based on simulated data and computed their Pearson correlation. Further, we added different levels of Gaussian white noise with standard deviation $\sigma = [0.01, 0.05, 0.1, 0.2, 0.3]$ to the simulated data to examine parameter recovery. For each noise level, we show the parameter estimates in the left two graphs. In the right two graphs, we plot the parameter estimates based on simulated data against the parameter values used to simulate the data. (**A**) Noise $\sigma = 0.01$. (**B**) Noise $\sigma = 0.05$. (**C**) Noise $\sigma = 0.1$. (**D**) Noise $\sigma = 0.2$. (**E**) Noise $\sigma = 0.3$. (**F**) Empirically estimated noise ($\sigma$) of each subject. Each bar represents a subject's estimated noise level. (**G–H**) Impact of noise homoscedasticity on parameter estimation. (**G**) Empirically estimated residual standard deviation. Mean residual standard deviation (across subjects, black data points) in the five probability intervals, [0.0–0.2], [0.2–0.4], [0.4–0.6], [0.6–0.8], and [0.8–1.0], were 0.1015, 0.1296, 0.1987, 0.1929, and 0.2061, respectively. Error bars represent ±1 standard error of the mean. (**H**) Parameter recovery results assuming heteroscedastic noise. We performed parameter recovery using the empirically estimated, probability-dependent residual variance shown in (**G**) (the mean residual standard deviation estimates). Error bars represent ±1 standard error of the mean (n=30).

Second, we fitted the system-neglect model to the simulated data. Third, we computed the correlation across subjects between the estimated parameters and the parameter values we used to simulate data. Fourth, we repeated the above steps by adding independent white noise to the simulated data. Across different levels of noise, we found good parameter recovery (Pearson's $r$ for transition probability $r \geq 0.9533$ across different noise levels, Pearson's $r$ for signal diagnosticity $r \geq 0.9515$ across different noise levels; *Figure 3*). In addition, we found that the empirical results (*Figure 2C*) can be reproduced by the system-neglect model (*Figure 4*). That is, we used each subject's parameter estimates to compute the period-wise probability estimates according to the system-neglect model and used these probability estimates to compute and plot index of overreaction (*IO*). The patterns of *IO* based on the system-neglect model (*Figure 4C*) were very similar to those based on subjects' actual data (*Figure 2C*).

## fMRI results

We focus our fMRI analyses on addressing three questions. First, what are the brain regions that correlated with subjects' probability estimates of change and belief revision? Second, what are the neural representations for the computational variables contributing to these probability estimates? Third, how might neural responses in the identified brain areas be associated with under- and overreactions to change?

## Ventromedial prefrontal cortex and ventral striatum represent regime-shift probability estimates and belief revision

Our first analysis is aimed at identifying brain regions that represented our subjects' regime-shift estimation. To address this question, we used two behavioral measures, namely the period-by-period probability estimates of regime shift, $P_t$, and the change in $P_t$ between successive periods, $\Delta P_t$. $P_t$ can be regarded as the subjects' posterior probability estimates of regime shift, whereas $\Delta P_t$ captures the change in belief (belief revision) about regime shift in the presence of a new signal (see *Figure 5A* for an example of $P_t$ and $\Delta P_t$).

For $P_t$, we found that the ventromedial prefrontal cortex (vmPFC) and ventral striatum correlated with this behavioral measure of subjects' belief about change. In addition, many other brain regions, including the motor cortex, central opercular cortex, insula, occipital cortex, and the cerebellum also significantly correlated with $P_t$. (*Figure 5B*; clusters in blue). For $\Delta P_t$, we also found that the vmPFC and ventral striatum were associated with regime shift belief revision (*Figure 5B*; clusters in orange). See GLM-1 in *Materials and methods*, *Figure 5—figure supplement 1*, and *Supplementary file 1*, *Supplementary file 2*, *Supplementary file 3*, respectively, for significant clusters of activation using Gaussian random field theory, permutation test on threshold-free-cluster-enhancement statistic, and permutations test on cluster-extent statistic. While many brain regions correlated with regime-shift probability estimates $(P_t)$, only the vmPFC and ventral striatum also correlated with belief revisions, $\Delta P_t$ (magenta clusters in *Figure 5B*).

Brain regions shown to correlate with regime-shift probability estimates, $P_t$, could be driven by motor response because larger estimates predominantly involved right-hand finger presses (see *Materials and methods* for details). To rule out motor confounds, we conducted two control experiments (Experiments 2 and 3) and performed two analyses. First, we examined the neural correlates of probability estimates ($P_t$ in GLM-1) in the control experiments (Experiments 2 and 3). Second, we compared the effect of $P_t$ (GLM-1) between the main experiment (Experiment 1) and the control experiments. In the first analysis, we found that in both control experiments, vmPFC and ventral striatum did not significantly correlate with probability estimates $P_t$ at the whole-brain level (in Experiment 2, no significant clusters of activation at the whole-brain level; see *Supplementary file 4* for Experiment 3). In the second analysis, we found that for both vmPFC and ventral striatum, the regression coefficient of $P_t$ was significantly different between Experiment 1 and Experiment 2 (*Figure 5C*) and between Experiment 1 and Experiment 3 (*Figure 5D*; also see *Supplementary file 5* and *Supplementary file 6*). In a separate, independent ROI analysis on vmPFC and ventral striatum, we also found the same results (*Figure 5EF*; see *Independent regions-of-interest (ROIs) analysis* in *Materials and methods* for details). Finally, we note that in GLM-1, we implemented an 'action-handedness' regressor to directly address the motor-confound issue, that higher probability estimates preferentially involved right-handed responses for entering higher digits. The action-handedness regressor was parametric,

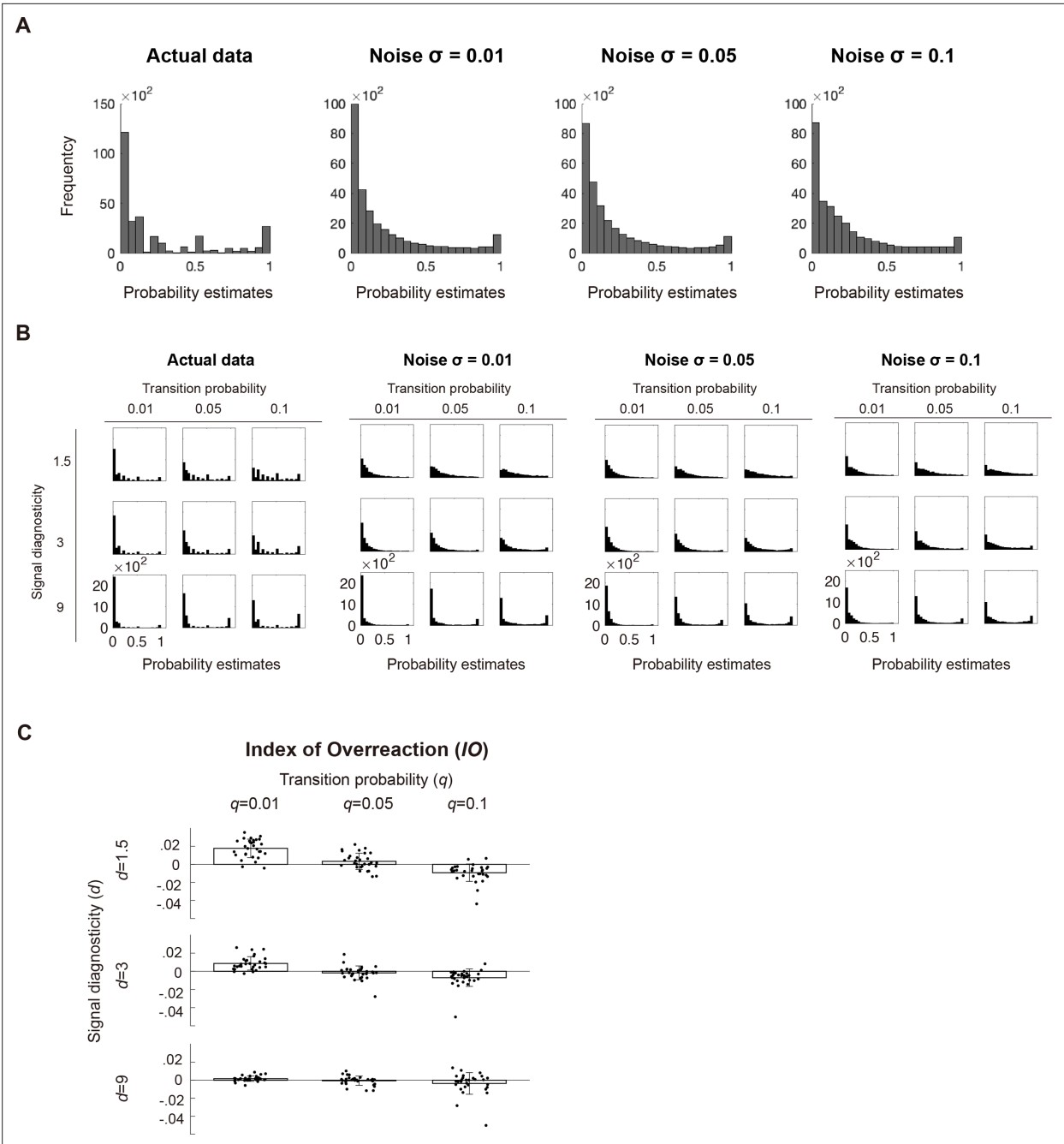

**Figure 4.** Probability estimates from the actual and simulated data. (**A**) Histogram of subjects' probability estimates collapsed across all conditions (left graph) and model-simulated probability estimates (system-neglect model) under three different noise levels (Noise $\sigma=0.01,\ 0.05,\ 0.1$). (**B**) Subjects' data are plotted as histograms separately for each condition. (**C**) System-neglect model can well-describe subjects' over- and underreactions to change. We fit the system-neglect model to each individual subject' probability estimates and used the resulting parameter estimates to compute each subject's probability estimates under the system-neglect model $\left(P_t^{SN}\right)$. We then used $P_t^{SN}$ to compute Index of Overreaction (*IO*). Here, *IO* was computed by subtracting belief revision predicted by the Bayesian model $\left(\Delta P_t^B = P_t^B - P_{t-1}^B\right)$ from belief revision estimated by system-neglect model $\left(\Delta P_t^{SN}=P_t^{SN} - P_{t-1}^{SN}\right)$. Formally, $IO=\Delta P_t^{SN} - \Delta P_t^B$. The mean *IO* (across all subjects; indicated by the bars) is plotted as a function of transition probability and signal diagnosticity. Data points in black represent individual subjects. Error bars represent ±1 standard error of the mean (n=30). The patterns of over- and underreactions here resembled those based on actual data (*Figure 2C*), suggesting that the system-neglect model can describe subjects' over- and underreactions well.

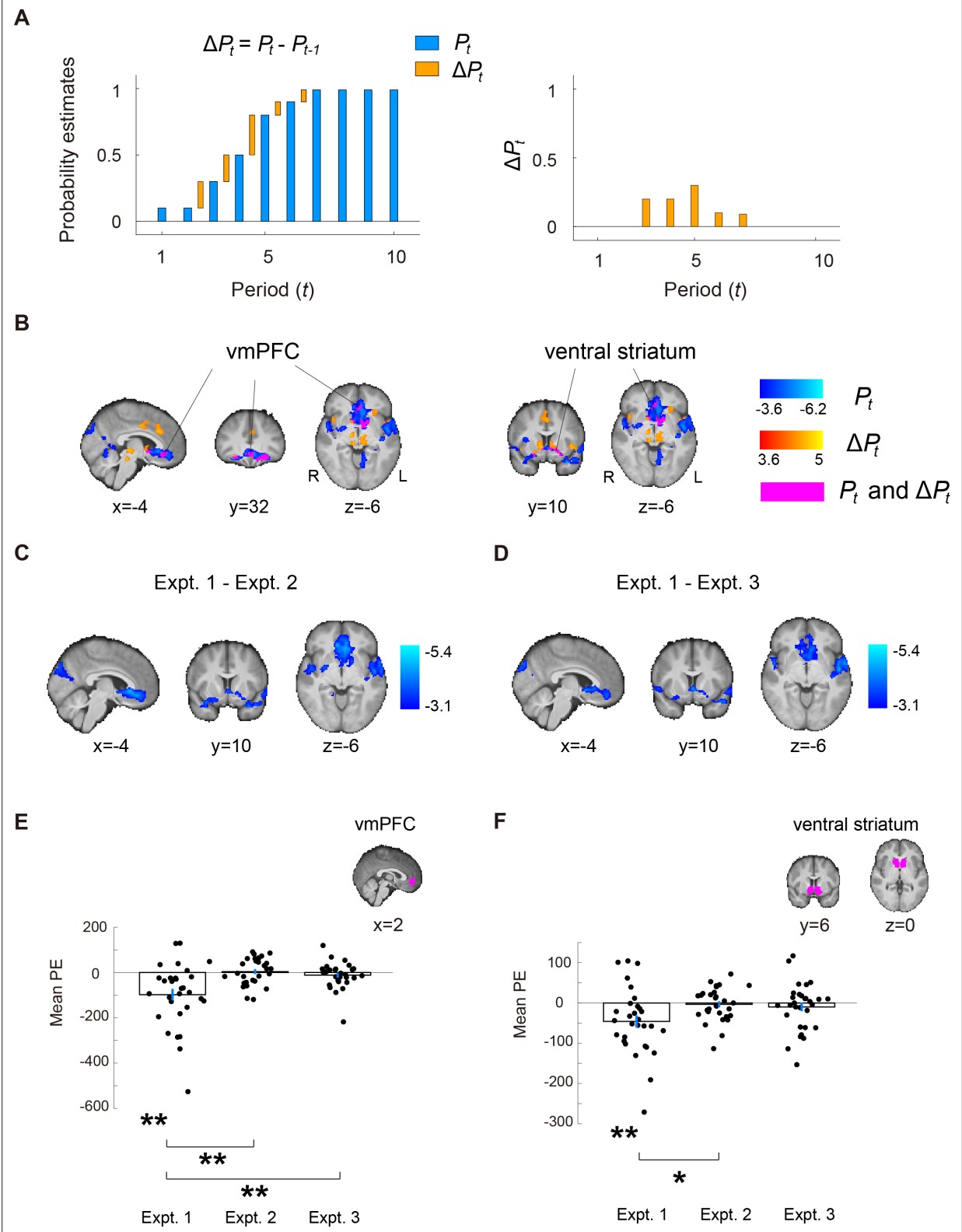

**Figure 5.** Neural representations for regime-shift probability estimates and belief revision. (**A**) An example. Belief revision (updating) is captured by the difference in probability estimates between two adjacent periods ($\Delta P_t$). The blue bars reflect the period-by-period probability estimates ($P_t$), while yellow bars depict $\Delta P_t$. (**B**) Whole-brain results (GLM-1) of the main experiment (Experiment 1) showing brain regions that significantly correlate with regime-shift probability estimates ($P_t$; clusters in blue) and the updating of beliefs about change ($\Delta P_t$; clusters in orange). Clusters in magenta

*Figure 5 continued on next page*

*Figure 5 continued*

indicate brain areas that correlate with both $P_t$ and $\Delta P_t$. (**C–D**) Between-experiment comparison of $P_t$. To rule out visual and motor confounds, we compared the $P_t$ contrast between the main experiment (Experiment 1) and two control experiments (Experiments 2 and 3). (**C**) Experiments 1 and 2 comparison. Whole-brain results of (Experiment 1 – Experiment 2) on the $P_t$ contrast. (**D**) Experiments 1 and 3 comparison. Whole-brain results of (Experiment 1 – Experiment 3) on the $P_t$ contrast. (**E–F**) Independent region-of-interest (ROI) analysis of vmPFC and ventral striatum on $P_t$ across the three experiments. For each subject and each ROI, we extracted the mean parameter estimates (PE) of the $P_t$ contrast from GLM-1. (**E**) vmPFC ROI. Experiment 1: One-sample $t$ test, $t(29) = -3.82$, $p<0.01$; Experiment 2: One-sample $t$ test, $t(29) = 0.36$, $p=0.71$; Experiment 3: One-sample $t$ test, $t(29) = -1.11$, $p=0.28$; Experiments 1 Experiment 2: two-sample $t$ test, $t(58) = -3.67$, $p<0.01$; Experiments 1 and 3: two-sample $t$ test, $t(58) = -3.12$, $p<0.01$. (**F**) Ventral striatum ROI. Experiment 1: $(29) = -3.06$, $p<0.01$; Experiment 2: $t(29) = 0.44$, $p=0.67$; Experiment 3: $t(29) = -0.93$, $p=0.36$; Experiments 1 and 2: $t(58) = -2.55$, $p=0.01$; Experiments 1 and 3: $t(58) = -1.95$, $p=0.06$. The * symbol indicates $p<0.05$ (two-tailed), and ** symbol indicates $p<0.01$ (two-tailed). Error bars represent ±1 standard error of the mean (n=30).

The online version of this article includes the following figure supplement(s) for figure 5:

**Figure supplement 1.** Whole-brain results of GLM-1 from the main experiment (Experiment 1) showing brain regions that significantly correlate with regime-shift probability estimates ($P_t$; clusters in blue in Panel **A**) and belief revision ($\Delta P_t$; clusters in orange in Panel **B**).

coding –1 if both finger presses involved the left hand (e.g. a subject pressed '23' as her probability estimate when seeing a signal), 0 if using one left finger and one right finger (e.g. '75'), and 1 if both finger presses involved the right hand (e.g. '90'). Taken together, these results ruled out motor confounds and suggested that vmPFC and ventral striatum represent subjects' probability estimates of change (regime shifts) and belief revision.

We further examined the robustness of $P_t$ and $\Delta P_t$ representations in vmPFC and ventral striatum in three follow-up analyses. In the first analysis, we implemented a GLM (GLM-2 in *Materials and methods*) that, in addition to $P_t$ and $\Delta P_t$, included various task-related variables contributing to $P_t$ as regressors. Specifically, to account for the fact that the probability of regime change increased over time, we included the *intertemporal prior* as a regressor in GLM-2. The intertemporal prior is the natural logarithm of the odds in favor of regime shift in the $t$-th period, $\ln\left(\frac{1-(1-q)^t}{(1-q)^t}\right)$, where $q$ is transition probability and $t=1,\ldots,10$ is the period (*Equation 1* in *Materials and methods*). It describes normatively how the prior probability of change increased over time regardless of the signals (blue and red balls) the subjects saw during a trial. Including it along with $P_t$ would clarify whether any effect of $P_t$ can otherwise be attributed to the intertemporal prior. We found that the results of $P_t$ and $\Delta P_t$ in the vmPFC and ventral striatum in GLM-2 were identical to those in GLM-1 (*Figure 6*): *Figure 6A* was meant to depict the results in slices identical to those shown in *Figure 5B* for results based on GLM-1. For slice-by-slice results, see *Figure 5—figure supplement 1* for results based on GLM-1 and *Figure 6—figure supplement 1* for GLM-2. For Tables of activations, see *Supplementary file 1*, *Supplementary file 2*, *Supplementary file 3* for GLM-1 and *Supplementary file 7*, *Supplementary file 8*, *Supplementary file 9* for GLM-2. In a separate, independent region-of-interest (ROI) analysis of vmPFC and ventral striatum (*Figure 6BC*; see *Independent regions-of-interest (ROIs) analysis* in *Materials and methods* for details), we further compared the effect of both $P_t$ and $\Delta P_t$ between GLM-1 and GLM-2. For $P_t$, the difference between GLM-1 and GLM-2 was significant in vmPFC but not in ventral striatum (paired t-test, $t(29) = -2.21$, $p=0.04$ in vmPFC, $t(29) = -0.85$, $p=0.40$ in ventral striatum), while the effect of $P_t$ from GLM-1 (one-sample t-test, $t(29) = -3.82$, $p < .01$ in vmPFC; $t(29) = -3.06$, $p < .01$ in ventral striatum) and GLM-2 was significant (one-sample t-test, $t(29) = -2.69$, $p=.01$ in vmPFC; $t(29) = -2.50$, $p=.02$ in ventral striatum). The significant difference in vmPFC between GLM-1 and GLM-2 suggested that the inclusion of intertemporal prior and other task-related regressors in GLM-2 did change the result of $P_t$ in vmPFC. However, vmPFC activity in both GLMs significantly correlated with $P_t$, suggesting that $P_t$ representations in vmPFC were present with and without the inclusion of intertemporal prior and other task-related regressors. For $\Delta P_t$, the difference between GLM-1 and GLM-2 was not significant (paired t-test, $t(29) = -0.22$, $p=0.83$ in vmPFC; $t(29) = 0.51$, $p=0.61$ in ventral striatum), while the effect of $\Delta P_t$ from GLM-1 (one-sample t-test, $t(29) = 3.12$, $p < .01$ in vmPFC; $t(29) = 4.17$, $p < .01$ in ventral striatum) and GLM-2 was significant (one-sample t-test, $t(29) = 2.92$, $p < .01$ in vmPFC; $t(29) = 3.59$, $p < .01$ in ventral striatum). For the intertemporal prior, activity in both vmPFC and ventral striatum did not correlate significantly with the intertemporal prior (one-sample t-test, $t(29) = 0.07$, $p=0.95$ in vmPFC; $t(29) = -0.53$, $p=0.60$ in ventral striatum). All the t-tests described above were two-tailed. Taken together, these results suggest that

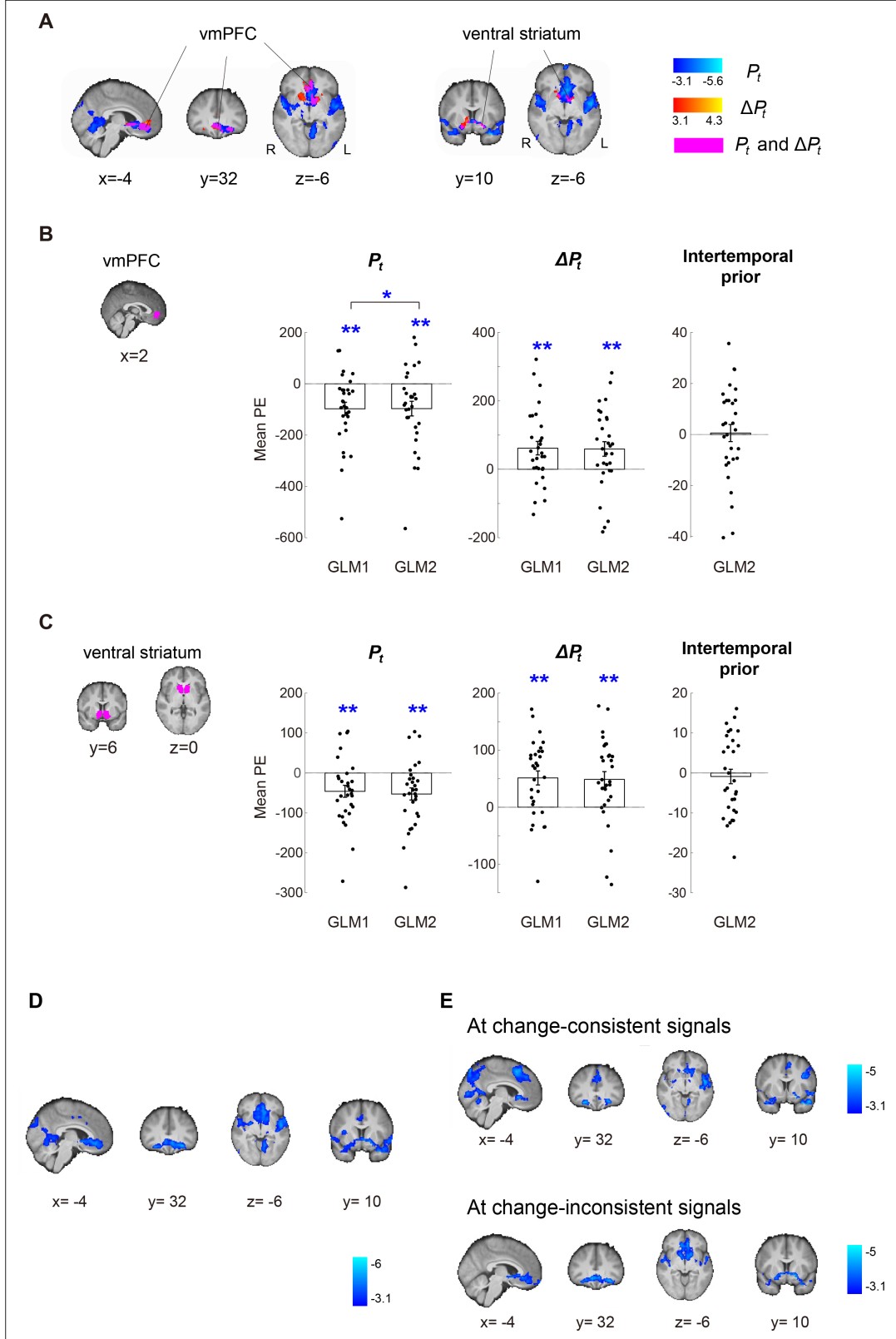

**Figure 6.** Robustness of neural representations for regime-shift probability estimates and belief revision in the vmPFC and ventral striatum. (**A**) Whole-brain results (GLM-2) of the main experiment (Experiment 1) showing brain regions that correlate with regime-shift probability estimates ($P_t$; clusters in blue) and the updating of beliefs about change ($\Delta P_t$; clusters in orange). Clusters in magenta represent brain areas that correlate with both $P_t$

*Figure 6 continued on next page*

*Figure 6 continued*

and $\Delta P_t$. (**B–C**) Independent region-of-interest (ROI) analysis of vmPFC and ventral striatum. We compared the effect of $P_t$ and $\Delta P_t$ estimated from GLM-1 with GLM-2, which differed on whether various task-related regressors contributing to $P_t$, especially the intertemporal prior, were included in the model. For a given ROI and a given regressor ($P_t$, $\Delta P_t$, or the intertemporal prior), we extracted the corresponding mean parameter estimates (PEs; averaged across voxels within the ROI) from each subject separately and plotted them. The bar height represents the mean across subjects. Each data point in black represents a single subject. Error bars represent ±1 standard error of the mean (n=30). (**B**) vmPFC results. (**C**) Ventral striatum results. (**D**) Whole-brain results of activity that significantly correlated with the subjects' log odds estimates of regime shift, $\ln\left(P_t/\left(1-P_t\right)\right)$. In this analysis, we replaced the parametric regressor of $P_t$ with the log odds of regime shifts in GLM-1. Familywise error-corrected at $p < 0.05$ using Gaussian random field theory with a cluster-forming threshold $z > 3.1$. (**E**) Whole-brain results of $P_t$ at change-consistent and change-inconsistent signals. We estimated the effect of $P_t$ separately at change-consistent (blue) and change-inconsistent (red) signals. The model was identical to GLM-1 except that we implemented R1-R5 in GLM-1 separately for change-consistent and change-inconsistent signals. Familywise error-corrected at $p < 0.05$ using Gaussian random field theory with a cluster-forming threshold $z > 3.1$.

The online version of this article includes the following figure supplement(s) for figure 6:

**Figure supplement 1.** Whole-brain results of GLM-2 from the main experiment (Experiment 1) showing brain regions that significantly correlate with regime-shift probability estimates ($P_t$; clusters in blue in Panel **A**) and belief revision ($\Delta P_t$; clusters in orange in Panel **B**).

vmPFC and ventral striatum represented $P_t$ and $\Delta P_t$ regardless of whether the intertemporal prior and other task-related regressors contributing to $P_t$ were included in the GLM. We also did not find that vmPFC and ventral striatum to represent the intertemporal prior. In the second analysis, we implemented a GLM that replaced $P_t$ with the log odds of $P_t$, $\ln\left(P_t/\left(1-P_t\right)\right)$ (*Figure 6D*). In the third analysis, we implemented a GLM that examined $P_t$ separately on periods when change-consistent (blue balls) and change-inconsistent (red balls) signals appeared (*Figure 6E*). Each of these analyses showed significant correlation with $P_t$ in vmPFC and ventral striatum, further establishing the robustness of the $P_t$ findings.

## A frontoparietal network represents key variables for estimating regime shifts

Our second analysis is aimed at identifying brain regions that represented key variables contributing to regime-shift estimation. Guided by our theoretical framework and computational models, we focused on two variables, the interaction between signals and signal diagnosticity and intertemporal prior probability of change (GLM-2 in *Materials and methods*) to examine these effects.

Our theoretical framework makes two fundamental predictions. First, a signal should be weighted differently depending on signal diagnosticity, that is a blue ball is stronger evidence for change in a highly diagnostic environment (e.g. $d$=9) than a system in which the red and blue regimes are very similar (e.g. $d$=1.5). To capture the interaction between signals and signal diagnosticity, we code a blue signal as 1 and a red signal as –1 and multiply the signal code ($s$=1 or –1) by the natural logarithm of signal diagnosticity, $\ln\left(d\right)$ (two examples are shown in *Figure 7A*). We term this interaction, $s \times \ln\left(d\right)$, the strength of evidence in favor of change or *strength of change evidence* for short. The Bayesian model, as described in *Materials and methods*, critically depends on $d^s$, computing posterior odds by multiplying prior odds by the likelihood ratio. Thus, the log posterior odds were calculated from both the prior odds and $s \times \ln\left(d\right)$ .. At the whole-brain level, we found that a frontoparietal network including the dorsal medial prefrontal cortex (dmPFC), lateral prefrontal cortex (bilateral inferior frontal gyrus, IFG), and the posterior parietal cortex (bilateral intraparietal sulcus, IPS) represented $s \times \ln\left(d\right)$ (*Figure 7A*). These brain regions overlap with what is commonly referred to as the frontoparietal control network (*Buckner et al., 2013*; *Seeley et al., 2007*; *Yeo et al., 2011*). Among them, dmPFC sits in the vicinity of the dorsomedial frontal cortex (DMFC) shown to represent change probability and uncertainty about change in reinforcement learning (*McGuire et al., 2014*).

The second prediction our theoretical framework offers concerns the prior probability of a regime shift over time. Specifically, the Bayesian model predicts that the prior probability should increase over time (see two examples in *Figure 7B*), with the *intertemporal prior*, in log odds terms, defined as the natural logarithm of the odds in favor of regime shift in the $t$-th period, $\ln\left(\frac{1-\left(1-q\right)^t}{\left(1-q\right)^t}\right)$, where

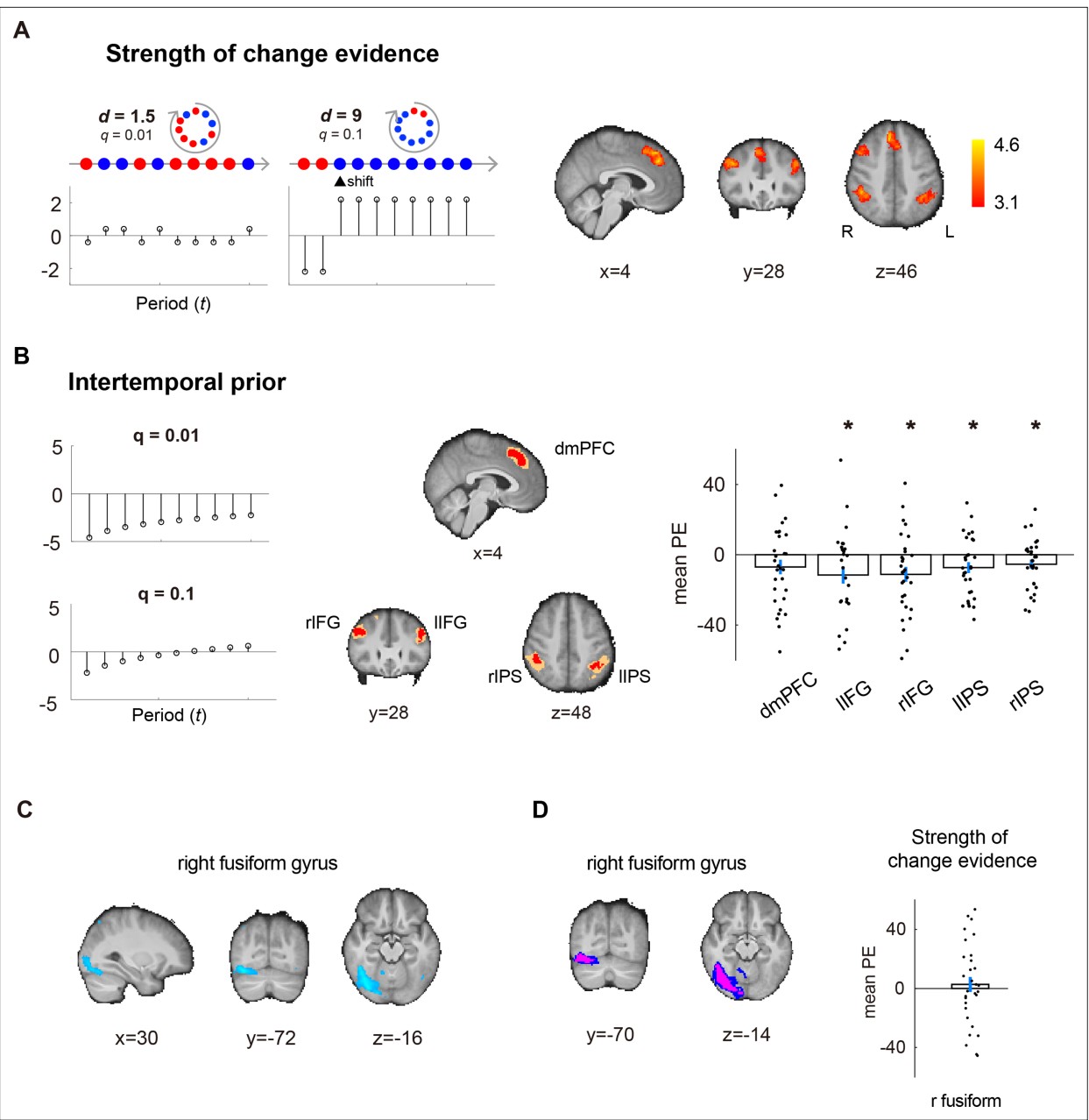

**Figure 7.** A frontoparietal network represents key variables for regime-shift estimation. (**A**) Variable 1: strength of change evidence measured by the interaction between signal diagnosticity and signal. Left: two examples of the interaction between signal diagnosticity (**d**) and signal (**s**), where a change-consistent (blue) signal is coded as 1 and a change-inconsistent (red) signal is coded as –1. The x-axis represents the time periods, from the first to the last period, in a trial. The y-axis represents the interaction, $\ln(d) \times s$. Right: whole-brain results showing brain regions in a frontoparietal network that significantly correlated with $\ln(d) \times s$. (**B**) Variable 2: intertemporal prior probability of change. Two examples of intertemporal prior are shown on the left graphs. To examine the effect of the intertemporal prior, we performed independent region-of-interest analysis (leave-one-subject-out, LOSO) on the brain regions identified to represent strength of change evidence. Due to the LOSO procedure, individual subjects' ROIs (a cluster of contiguous voxels) would be slightly different from one another. To visualize such differences, we used the red color to indicate voxels shared by all individual subjects' ROIs, and orange to indicate voxels by at least one subject's ROI. The ROI analysis examined the regression coefficients (mean PE) of intertemporal prior. The * symbol indicates $p < 0.05$, ** indicates $p < 0.01$. dmPFC: dorsomedial prefrontal cortex; lIPS: left intraparietal sulcus; rIPS: right intraparietal sulcus; lIFG: left inferior frontal gyrus; rIFG: right inferior frontal gyrus. (**C**) Whole-brain results of the intertemporal prior of regime shift. (**D**) Using the intertemporal prior ROI (left graph: magenta indicates voxels shared by the LOSO ROI of all subjects; blue indicates voxels of LOSO ROI of at least one subject) to examine the regression coefficients of the strength of change evidence, $\ln(d) \times$ signal. The mean parameter estimates (mean PE), i.e., regression coefficient, was not significantly different from 0 (one-sample t test, $t(29) = 0.54$, $p=0.59$, two-tailed). Error bars represent ±1 standard error of the mean.

$q$ is transition probability and $t=1,\ldots,10$ is the period (*Equation 1* in *Materials and methods*). With independent (leave-one-subject-out, LOSO) ROI analysis, we examined whether brain regions in the frontoparietal network (shown to represent strength of change evidence) correlated with intertemporal prior and found that all brain regions, with the exception of dmPFC, in the frontoparietal network correlated with the intertemporal prior (*Figure 7B*; dmPFC: $t(29) = -1.69$, $p=0.10$; left IFG: $t(29) =2.20$, $p=0.04$; right IFG: $t(29) = -2.64$, $p=0.01$; left IPS: $t(29) = -2.35$, $p=0.03$; right IPS: $t(29) = -2.07$, $p=0.05$). By contrast, brain regions that represented the intertemporal prior, which we found to be in the right fusiform cortex in the occipitotemporal regions, did not correlate with the strength of change evidence (*Figure 7C and D*).

Finally, we emphasize that these effects—the strength of change evidence and intertemporal prior—cannot be otherwise attributed to probability estimates ($P_t$) or belief revision ($\Delta P_t$) because both $P_t$ and $\Delta P_t$ were included in GLM-2 where these effects were examined. Taken together, these results suggest that the frontoparietal network is critically involved in representing the two key variables for estimating regime shifts, strength of change evidence and intertemporal prior. See *Supplementary file 7*, *Supplementary file 8*, *Supplementary file 9* for information about significant clusters of activation for the strength of change evidence, intertemporal prior, $P_t$, and $\Delta P_t$ from GLM-2 using Gaussian random field theory (*Supplementary file 7*), permutation test on threshold-free-cluster-enhancement (TFCE) statistic (*Supplementary file 8*), and permutation test on cluster-extent statistic (*Supplementary file 9*).

## Under- and overreactions are associated with selectivity and sensitivity of neural responses to system parameters

The system-neglect hypothesis posits that under- and overreactions arise from a lack of sensitivity to the system parameters. We can measure individual subjects' sensitivity to system parameters using behavioral data (subjects' probability estimates). Meanwhile, we can also measure sensitivity using neural data. In the following analysis, we examined whether there is a match between the behavioral and neural measures of sensitivity to the system parameters. This would allow us to examine, through the system-neglect framework, whether sensitivity in neural responses to the system parameters are associated with under- and overreactions to change.

We focused on the vmPFC-striatum network and frontoparietal network, as they were shown to be involved in regime-shift detection (*Figures 5–7*). We examined whether these brain networks show selective preference for a particular system parameter, which we refer to as *parameter selectivity*. We also asked whether parameter selectivity is signal-dependent, i.e., different for signals consistent with change (blue signals) or inconsistent with change (red signals).

We started by defining a behavioral measure of sensitivity to the system parameter. To visualize this measure, we consider two extreme decision makers, a Bayesian and someone who reacts to signals identically across all systems, which we term *complete neglect*. In *Figure 8A* (left graph), we use signal diagnosticity ($d$) to illustrate the pattern of these two decision makers. The vertical axis is $\beta \ln(d)$ and the horizontal axis is the signal-diagnosticity level ($d$), where $\beta$ is the weighting parameter on signal diagnosticity in the system-neglect model. A Bayesian (open circles) does not overweight or underweight $d$, and thus $\beta=1$. We can then define the *Bayesian slope* by regressing $\beta_i \ln(d_i)$ against $\ln(d_i)$. In this formulation, the Bayesian slope is 1 and it reflects the sensitivity of a Bayesian decision maker to signal diagnosticity. On the other hand, a complete-neglect decision maker is unresponsive to signal diagnosticity, that is $\beta_1 \ln(d_1) = \beta_2 \ln(d_2) = \beta_3 \ln(d_3)$. Hence, the *complete-neglect slope* should be 0. These two slopes, the Bayesian slope and the complete-neglect slope, provide the boundaries for system neglect. For each subject, we computed $\beta_i \ln(d_i)$ at each $d_i$ level, where $\beta_i$ is the estimate for diagnosticity $d_i$ fitted to the system-neglect model (see $\beta_i$ in *Figure 2D*). We then estimated each subject's *behavioral slope* (to distinguish it from the *neural slope* reported later) and used it as a behavioral measure of sensitivity to signal diagnosticity.

For each subject, we estimated two behavioral slopes, one for $d$, the signal diagnosticity (top row in *Figure 8A*), and the other for $q$, the transition probability (bottom row in *Figure 8A*). The right graphs in *Figure 8A* show the behavioral slope for each of the 30 subjects (top: signal diagnosticity; bottom: transition probability). For signal diagnosticity, 28 out of 30 subjects' behavioral slopes were within the boundaries. For transition probability, 27 out of 30 subjects' behavioral slopes were within the boundaries (between complete neglect and Bayesian). One subject's (subject 6) behavioral slope

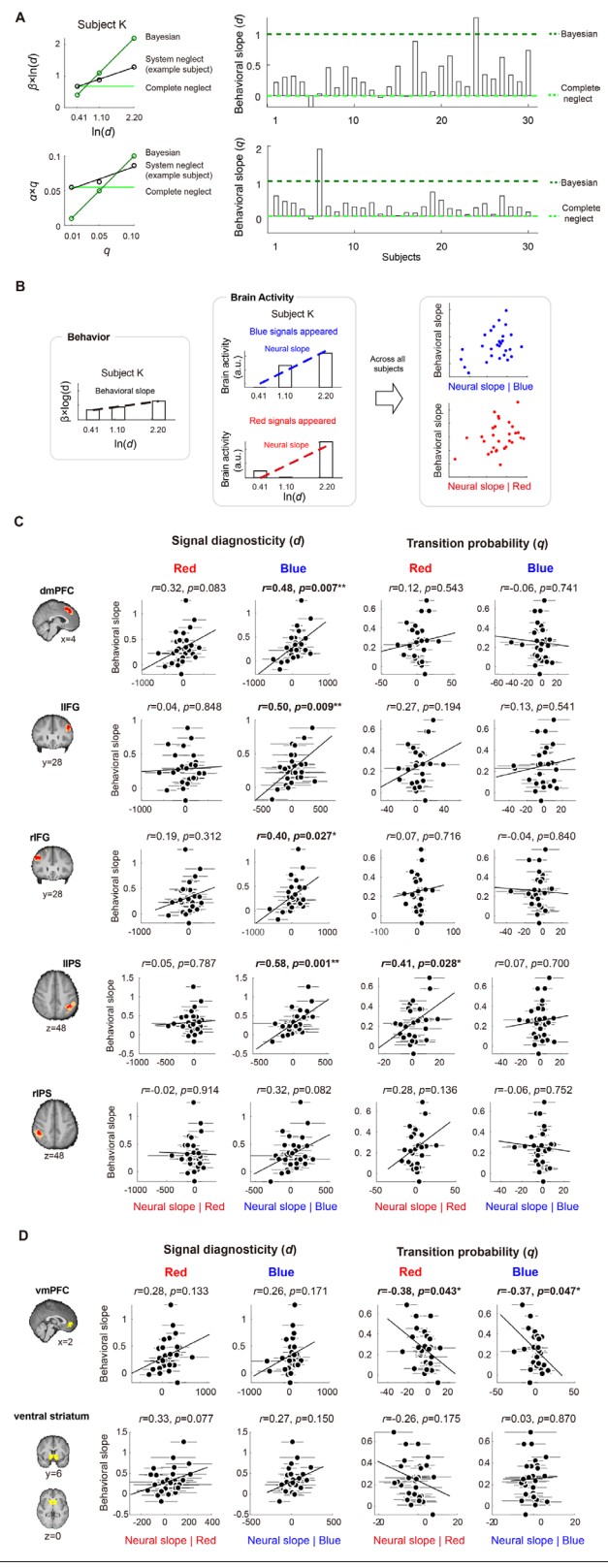

**Figure 8.** Estimating and comparing neural measures of sensitivity to system parameters with behavioral measures of sensitivity. (**A**) Behavioral measures of sensitivity to system parameters. For each system parameter, we plot the subjectively weighted system parameter against the system parameter level (top row: signal diagnosticity; bottom row: transition probability). For each subject and each system parameter, we estimated the slope (how

*Figure 8 continued*

the subjectively weighted system parameter changes as a function of the system parameter level) and used it as a behavioral measure of sensitivity to the system parameter (behavioral slope). We also show a Bayesian (no system neglect) decision maker's slope (dark green) and the slope of a decision maker who completely neglects the system parameter (in light green; the slope would be 0). A subject with stronger neglect would have a behavioral slope closer to complete neglect. (**B**) Comparison of behavioral and neural measures of sensitivity to the system parameters. To estimate neural sensitivity, for each subject and each system parameter, we regressed neural activity of a ROI against the parameter level and used the slope estimate as a neural measure of sensitivity to that system parameter (neural slope). We also estimated the neural slope separately for change-consistent signal periods (when the subject saw a blue signal) and change-inconsistent signal periods. We computed the Pearson correlation coefficient (*r*) between the behavioral slope and the neural slope and used it to statistically test whether there is a match between the behavioral and neural slopes. (**C**) The frontoparietal network selectively represented individuals' sensitivity to signal diagnosticity (left two columns), but not transition probability (right two columns). Further, neural sensitivity to signal diagnosticity (neural slope) correlated with behavioral sensitivity (behavioral slope) only when a signal in favor of potential change (blue) appeared: all the regions except the right IPS showed statistically significant match between the behavioral and neural slopes. By contrast, sensitivity to transition probability was not represented in the frontoparietal network. (**D**) The vmPFC selectively represented individuals' sensitivity to transition probability ($r = -0.38$, $p=0.043$ for change-inconsistent signals; $r = -0.37$, $p=0.047$ for change-consistent signals), but not signal diagnosticity ($r=0.28$, $p=0.13$ for change-inconsistent signals; $r=0.26$, $p=0.17$ for change-consistent signals). The ventral striatum did not show selectivity to either transition probability or signal diagnosticity. Error bars represent ±1 standard error of the mean.

for *q* (transition probability) was approximately 2 and clearly outside the boundaries. This subject's data were excluded for further analysis of *q* (the right two columns in *Figure 8C and D*).

We found that, across subjects, system neglect was unique from either Bayesian or complete neglect. Subjects' sensitivity to transition probability, as captured by the behavioral slope in *Figure 8A*, deviated significantly from the Bayesian slope (comparing subjects' slope with 1, $t(29) = -10.8$, $p<.01$, two-tailed) and from complete neglect slope (comparing subjects' slope with 0, $t(29) = 4.8$, $p<.01$, two-tailed). For signal diagnosticity, subjects' sensitivity to signal diagnosticity was also significantly different from both Bayesian ($t(29) = -12.5$, $p<.01$, two-tailed) and complete neglect ($t(29) = 6.1$, $p<.01$, two-tailed).

However, subjects were closer to complete neglect than to the Bayesian. We tested this by examining whether the behavioral slope, $\gamma$, was significantly greater or smaller than 0.5, the midpoint between complete neglect (slope of 0) and Bayesian (slope of 1). $\gamma - .5 > 0$ indicates that subjects' behavior was in closer alignment with Bayesian. By contrast, $\gamma - .5 < 0$ implies behavior closer to complete neglect. We found that, for both transition probability and signal diagnosticity, the behavioral slope was closer to complete neglect than to Bayesian (transition probability: $t(29) = -2.97$, p < 0.01; signal diagnosticity: $t(29) = -3.23$, p < 0.01, two-tailed). Together, these results suggested that, while subjects did respond to the system parameters in regime-shift estimation in the correct direction predicted by the Bayesian model, their sensitivity to the system parameters was closer to complete neglect than to normative Bayesian.

For the neural data, we defined a neural measure of sensitivity to the system parameters by estimating how neural responses change as a function of those parameters. Using the signal diagnosticity parameter as an example, for each subject and each ROI separately, we regressed average brain activity at each diagnosticity level against $\ln(d)$. The slope estimate, termed the *neural slope*, from the linear regression gave us a neural measure of sensitivity to signal diagnosticity. To investigate whether the neural sensitivity was signal-dependent, that is neural sensitivity in response to the change-consistent signals (blue signals) was different from the change-inconsistent signals (red signals), we separately estimated the neural slope in response to blue and red signals.

After obtaining both the behavioral and neural measures of sensitivity to the system parameters, we then computed the Pearson correlation coefficient between them. We found that the vmPFC-striatum network and frontoparietal network showed clear dissociations in how they corresponded with the system parameters. First, the frontoparietal network represented individual subjects' sensitivity to signal diagnosticity (left two columns in *Figure 8C*), but not transition probability (right two columns in *Figure 8C*). Notably, patterns of parameter selectivity were remarkably consistent across brain regions in the frontoparietal network: when change-consistent signals (blue signals) appeared,

the neural measure of sensitivity from all brain regions in the frontoparietal network except the right IPS significantly correlated with the behavioral measure of sensitivity (second column from the left in *Figure 8C*; dmPFC: $r=0.48$, $p=0.007$; lIFG: $r=0.5$, $p=0.009$; rIFG: $r=0.4$, $p=0.027$; lIPS: $r=0.58$, $p=0.001$; rIPS108143: $r=0.32$, $p=0.082$). By contrast, when change-inconsistent signals (red signals) appeared, all regions within the frontoparietal network did not significantly correlate with the behavioral measure of sensitivity (first column from the left in *Figure 8C*; dmPFC: $r=0.32$, $p=0.083$; lIFG: $r=0.04$, $p=0.848$; rIFG: $r=0.19$, $p=0.312$; lIPS: $r=0.05$, $p=0.787$; rIPS: $r=-0.02$, $p=0.914$). We further tested, for each brain region, whether the difference in correlation was significant using both parametric and nonparametric tests (see *Parametric and nonparametric tests for difference in correlation coefficients* in *Materials and methods*). The results were identical. In the parametric test, we used the Fisher $z$ transformation to transform the correlation coefficients to the $z$ statistic. Since these correlation coefficients were not independent, we compared them using the test developed in *Meng et al., 1992* (see *Materials and methods*). We found that among the five ROIs in the frontoparietal network, two of them, namely the left IFG and left IPS, the difference in correlation was significant (one-tailed z test; left IFG: $z=1.8908$, $p=0.0293$; left IPS: $z=2.2584$, $p=0.0049$). For the remaining three ROIs, the difference in correlation was not significant (dmPFC: $z=0.9522$, $p=0.1705$; right IFG: $z=0.9860$, $p=0.1621$; right IPS: $z=1.4833$, $p=0.0690$). We chose one-tailed test because we already know the correlation under change-consistent signals was significantly greater than 0. In the nonparametric test, we performed nonparametric bootstrapping to test for the difference in correlation. We referred to the correlation between neural and behavioral sensitivity at change-consistent (blue) signals as $r_{blue}$, and that at change-inconsistent (red) signals as $r_{red}$. Consistent with the parametric tests, we also found that the difference in correlation was significant in left IFG and left IPS (left IFG: $r_{blue} - r_{red}=0.46$, $p=0.0496$; left IPS: $r_{blue} - r_{red}=0.5306$, $p=0.0041$), but was not significant in dmPFC, right IFG, and right IPS (dmPFC: $r_{blue} - r_{red}=0.1634$, $p=0.1919$; right IFG: $r_{blue} - r_{red}=0.2123$, $p=0.1681$; right IPS: $r_{blue} - r_{red}=0.3434$, $p=0.0631$). In summary, we found that neural sensitivity to signal diagnosticity measured at change-consistent signals significantly correlated with individual subjects' behavioral sensitivity to signal diagnosticity. By contrast, neural sensitivity to signal diagnosticity measured at change-inconsistent signals did not significantly correlate with behavioral sensitivity. The difference in correlation, however, was statistically significant in some (left IPS and left IFG) but not all brain regions within the frontoparietal network.

Second, in contrast to the frontoparietal network, vmPFC in the vmPFC-striatum network showed the opposite pattern of parameter selectivity: vmPFC selectively represented individual subjects' sensitivity to transition probability (right two columns in *Figure 8D*), but not to signal diagnosticity (left two columns in *Figure 8D*). Selectivity in vmPFC was not signal-dependent: regardless of change-consistent (blue) or change-inconsistent (red) signals, neural sensitivity to transition probability in vmPFC represented individual subjects' sensitivity to transition probability ($r=-0.38, p=0.043$ for change-inconsistent signals; $r=-0.37, p=0.047$ for change-consistent signals). By contrast, the ventral striatum did not show selectivity to either the transition probability or signal diagnosticity (transition probability: $r=-0.26, p=0.175$ for change-inconsistent signals; $r=0.03, p=0.870$ for change-consistent signals; signal diagnosticity: $r=0.33, p=0.077$ for change-inconsistent signals; $r=0.27, p=0.150$ for change-consistent signals). In summary, these results suggest that vmPFC selectively represented individuals' sensitivity to transition probability, whereas the frontoparietal network selectively represented individuals' sensitivity to signal diagnosticity.

## Incorporating signal dependency into system-neglect model led to better models for regime-shift detection

The neural findings on signal dependency (*Figure 8*) point to the possibility that participants might respond to the system parameters differently when facing change-consistent and change-inconsistent signals. This led us to ask whether building signal dependency into the system-neglect model would be a better model choice for subjects' behavioral data (probability estimates of regime shift) than the original system-neglect model. To examine this question, we built and fit three new versions of the system-neglect (SN) model (see *Supplementary file 10* for model-fitting summary) and compared them with the original model (SN-original; see *Supplementary file 11* for summary of statistical tests for model comparison). In the signal-dependent $\beta$ system-neglect model (SN-SigDep-$\beta$ model), we estimated the $\beta$ parameters separately at change-consistent and change-inconsistent signals. As a result, in this model there were 6 $\beta$ parameters—three for change-consistent signals to model each of

the three levels of signal diagnosticity and three for change-inconsistent signals—and three $\alpha$ parameters that modeled each of the three levels of transition probability without distinguishing between change-consistent and change-inconsistent signals. In the signal-dependent $\alpha$ system-neglect model (SN-SigDep-$\alpha$ model), we estimated the $\alpha$ parameters separately at change-consistent and change-inconsistent signals. As a result, in this model there were 6 $\alpha$ parameters (three for change-consistent signals and three for change-inconsistent signals) and three $\beta$ parameters. In the signal-dependent $\alpha$ and $\beta$ system-neglect model (SN-SigDep-$\alpha\beta$ model), we estimated both $\alpha$ and $\beta$ parameters separately at change-consistent and change-inconsistent signals (12 total parameters). Compared with SN-original, we found that SN-SigDep-$\beta$, SN-SigDep-$\alpha$, and SN-SigDep-$\alpha\beta$ qualitatively described subjects' behavioral data (belief revision, $\Delta P_t$) better (*Figure 9B–E*). Further, we found that estimating $\alpha$ separately at change-consistent and change-inconsistent signals (SN-SigDep-$\alpha$, *Figure 9D*) model improved model fits than estimating $\beta$ separately (SN-SigDep-$\beta$, *Figure 9C*), suggesting that subjects responded to transition probability differently when facing change-consistent and change-inconsistent signals more than to signal diagnosticity. Model comparison using Akaike Information Criterion (AIC) revealed that SN-SigDep-$\alpha\beta$ is the best model, followed by SN-SigDep-$\alpha$, SN-SigDep-$\beta$, and SN-original (*Figure 9F*). Together, these results suggest that participants showed system-neglect to both transition probability and signal diagnosticity and that they responded to these system parameters differently when facing change-consistent and change-inconsistent signals. In summary, signal dependency in response to system parameters is a new behavioral finding not reported in the original *Massey and Wu, 2005* study and is largely inspired by the neural sensitivity findings in the current study.

## Discussion

In this study, we investigated how humans detect changes in the environments and the neural mechanisms that contribute to how we might under- and overreact in our judgments. Combining a novel behavioral paradigm with computational modeling and fMRI, we discovered that sensitivity to environmental parameters that directly impact change detection is a key mechanism for under- and overreactions. This mechanism is implemented by distinct brain networks in the frontal and parietal cortices and in accordance with the computational roles they played in change detection. By introducing the framework in system neglect and providing evidence for its neural implementations, this study offered both theoretical and empirical insights into how systematic judgment biases arise in dynamic environments.

Regime shifts—the transition from one state of the world to another—are present in many daily situations, from the stock market (a change from the bull to the bear market) to the state of a pandemic. Detecting regime shifts can be challenging for at least two reasons. First, the signals we receive from the environments are often noisy. A signal in favor of potential change, for example a drop in pandemic cases, can either inform a true shift in regime or simply reflect noisy fluctuations. Second, the signals we receive reflect the volatility of the environment: while some environments are more prone to changes, others are not. To capture these two key features in regime-shift detection, we designed an fMRI task based on *Massey and Wu, 2005* where subjects made probability judgments about regime shifts and where we manipulated the signal diagnosticity and transition probability. Signal diagnosticity captures the level of noise inherent in the signals, while transition probability reflects the volatility of the environment. Replicating *Massey and Wu, 2005*, we found that overreactions to regime shifts take place when participants received noisy signals (low signal diagnosticity) but when the environments were stable (low transition probability). By contrast, when the signals are more precise but the environments were unstable, participants tended to underreact to changes. These results suggest system neglect—people respond primarily to signals and secondarily to the system that generates the signals (*Massey and Wu, 2005*).

At the neurobiological level, we found that regime-shift detection is jointly implemented by two networks, the vmPFC-striatum network and a frontoparietal network. The vmPFC-striatum network represented subjects' probability estimates of change and the revision of probability estimates in the presence of new signals (belief revision). By contrast, the frontoparietal network represented the strength of change evidence and intertemporal prior probability of change—two key variables contributing to probability estimation. Guided by the system-neglect framework, we found that under- and overreactions to change are closely associated with the sensitivity of these networks in response to the

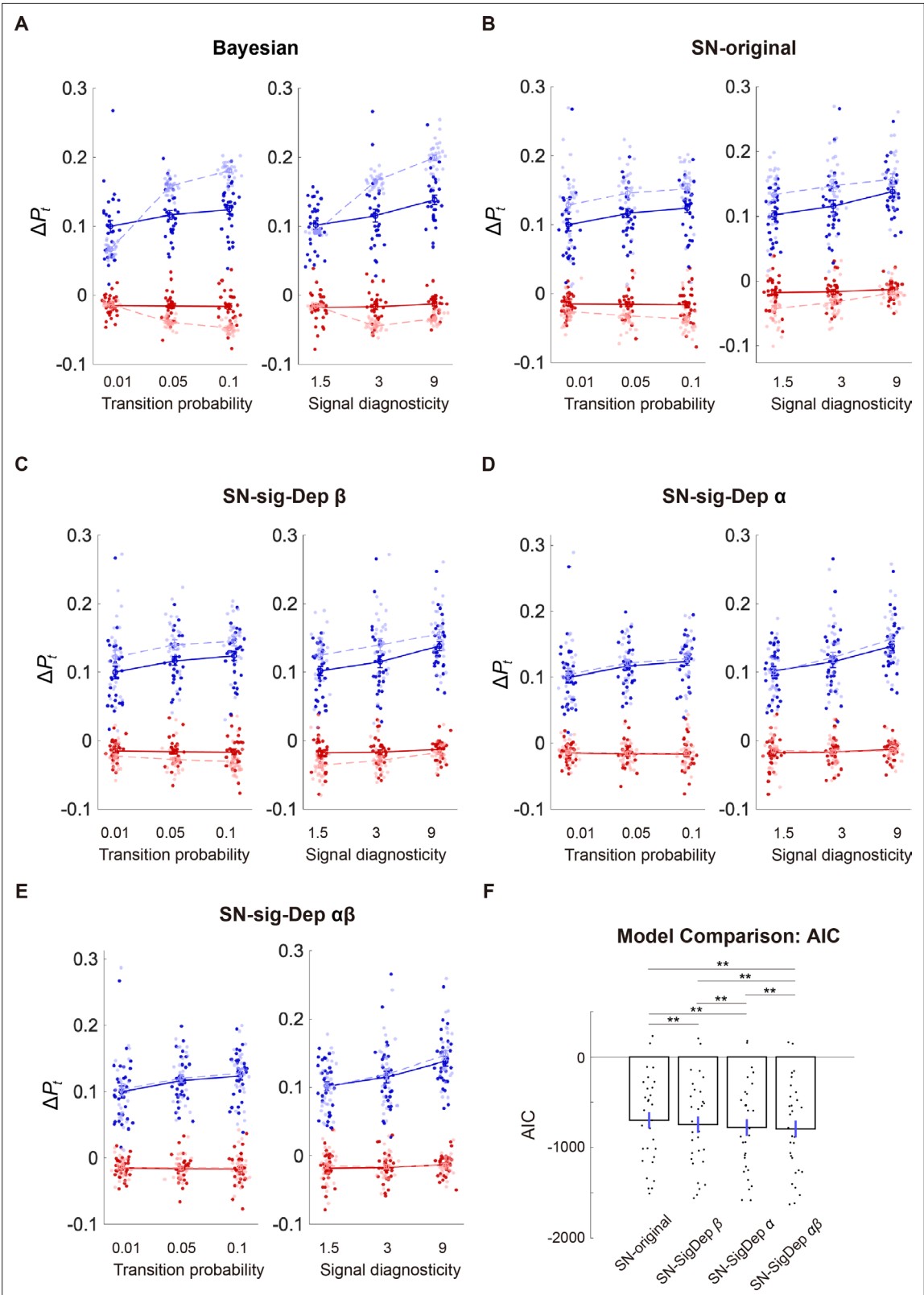

**Figure 9.** Model comparison. (**A–E**) Modeling results from five competing models. For each model, we plot subjects' belief revision ($\Delta P_t$) and the model-estimated $\Delta P_t$. Light-colored dots and dashed lines, respectively, represent the model-estimated $\Delta P_t$ at the individual and group levels. Dark-colored dots and solid lines indicate individual subjects' $\Delta P_t$ and group-averaged behavioral data, respectively. Blue indicates data and model estimates at change-consistent signals; Red indicates data and model estimates at change-inconsistent signals. (**A**) Bayesian model. (**B**) Original

*Figure 9 continued*

system-neglect model (SN-original). (**C**) Signal-dependent *β* system-neglect model (SN-SigDep-*β*). (**D**) Signal-dependent *α* system-neglect model (SN-SigDep-*α*). (**E**) Signal-dependent *α* and *β* system-neglect model (SN-SigDep-*αβ*). (**F**) Model comparison based on the Akaike Information Criterion (AIC). Lower AIC values indicate better models. The bars indicate group mean AIC (averaged across all subjects), while the black dots indicate individual subjects' AIC values. Error bars represent ±1 standard error of the mean (n=30). The * symbol indicates $p < 0.05$, ** indicates $p < 0.01$ (paired t-test; see **Supplementary file 11** for summary of statistical tests).

system parameters—transition probability and signal diagnosticity—that impact regime changes. In particular, the vmPFC represented individual subjects' sensitivity to transition probability, whereas the frontoparietal network represented sensitivity to signal diagnosticity. Together, these findings suggest that selectivity and sensitivity of neural responses to system parameters are key mechanisms that give rise to under- and overreactions.

Our work is closely related to the reversal-learning paradigm—the standard paradigm in neuroscience and psychology to study change detection (*Fellows and Farah, 2003*; *Izquierdo et al., 2017*; *O'Doherty et al., 2001*; *Schoenbaum et al., 2000*; *Walton et al., 2010*). In a typical reversal-learning task, human or animal subjects choose between two options that differ in the reward magnitude or probability of receiving a reward. Through reward feedback, the participants gradually learn the reward contingencies associated with the options and have to update knowledge about reward contingencies when contingencies are switched in order to maximize rewards. While a switch in reward contingencies can be regarded as a kind of regime shift, there are three major differences between the reversal-learning paradigm and our regime-shift task. The first difference is about learning. In the reversal-learning paradigm, the subjects must learn both the reward contingencies and the switches through experience. By contrast, in the regime-shift task, the subjects were explicitly informed about the makeup of different regimes and the transition probability. Therefore, participants do not need to learn about the different regimes and the transition probability through experience. The second difference is the kind of behavioral data collected. In our task, we asked the subjects to estimate the probability of change, whereas in the reversal-learning task, the subjects indicate their choice preferences. The third difference is on reward contingency. In the reversal-learning task, a change is specifically about change in reward contingencies, which is not the case in our task.

We believe that these major differences in task design led to three key insights into change detection from our study. The first insight is on over- and underreactions to change. At the behavioral level, we were able to identify situations that led to over- and underreactions. At the theoretical level, we were able to provide a systematic account for these over- and underreactions in the system-neglect hypothesis. Finally, at the neurobiological level, we were able to quantify the degree to which individual subjects neglected the system parameters and use these behavioral measures to unravel the neural mechanisms that give rise to over- and underreactions to change.

The second insight is on the brain networks associated with change detection. In particular, we were able to clarify whether the neural systems involved in change detection in the reversal-learning tasks are contingent on whether rewards are involved. Since the reversal-learning tasks are about learning the reward contingencies and the change in reward contingencies, it would be challenging to infer whether the neural implementations of change detection are dissociable from reward processing. Indeed, brain regions shown to be involved in the reversal-learning tasks, the OFC, mPFC, striatum, and amygdala, were also found to be highly involved in reward-related learning and value-based decision making. In the current study, unlike in reversal-learning paradigms, regimes were not defined by rewards (e.g. high reward-probability regime vs. low reward-probability regime in reversal learning paradigm). Therefore, estimating the probability of regime shifts in our task did not require considerations for change in reward contingencies. Our findings that vmPFC and ventral striatum represent probability estimates of change and belief revision therefore suggest that these brain regions might be part of a common pathway for change detection in general where changes in the state of the world do not have to be about changes in reward contingencies.

The third insight has to do with the impact of learning on change detection. Under the reversal-learning paradigm, it has been challenging to infer whether there exists a unique change-detection mechanism that is dissociable from reinforcement learning mechanisms. The way to make such inference is through theory, such as implementing a prior for state changes (*Bartolo and Averbeck, 2020*; *Costa et al., 2015*). Unlike the reversal-learning task, participants in our task did not have to learn

about the different regimes through experience. Without the confound of reinforcement learning, our results help clarify the roles of change detection on choice behavior by suggesting that independent of learning, there exists a specialized change-detection mechanism in the brain that impacts decision making. This mechanism involves the participation of the vmPFC-striatum network and the frontoparietal network, which partially overlap with the brain regions involved in reversal learning. However, it remains to be seen how learning interacts with change detection. Future investigations can address this question by combining the key features of both the reversal-learning paradigm and regime-shift paradigm.

Outside of the reversal-learning paradigm, previous fMRI studies that investigated learning and belief updating in dynamic environments where change takes place regularly had identified brain regions that represent perceived likelihood of change inferred from participants' choice behavior. *Payzan-LeNestour et al., 2013* identified that the posterior cingulate cortex, postcentral gyrus, middle temporal gyrus, hippocampus, and insula correlated with subjects' perceived likelihood of change in a multi-arm bandit task. *McGuire et al., 2014* found that subjective change-point probability was represented in a large posterior cluster including occipital, inferior temporal, and posterior parietal cortex. In addition, activity in dorsomedial frontal cortex, posterior cingulate cortex, superior frontal cortex, and anterior insula also positively correlated with change probability. Interestingly, both *McGuire et al., 2014* and our results found that the ventral striatum negatively correlated with probability estimates of change. This result suggested that the ventral striatum represents probability estimates of change irrespective of whether the task was based on a learning-based paradigm (*McGuire et al., 2014*) or a non-learning paradigm where information about task-related variables was explicitly revealed to the participants. Further, both McGuire et al. and our results found the involvement of the dorsomedial prefrontal cortex (dmPFC; or dorsomedial frontal cortex in McGuire et al.) in change detection. Our results further suggest that dmPFC is specialized in weighing the strength of change evidence and represents individual subjects' sensitivity to signal diagnosticity, both of which played important roles in contributing to the over- and underreactions to change.

How might our results relate to value-based decision making? In previous studies, vmPFC had been implicated to dynamically track financial risks that carry potential monetary gains or losses. To understand dynamic computations of risk, *Schonberg et al., 2012* used a Balloon Analog Risk Task (BART) where subjects decide whether to inflate a simulated balloon through successive pumps for the potential to win larger gains or incur larger losses (if the balloon explodes), or to cash out before the balloon explodes. They found that vmPFC activity decreased as subjects pumped and expanded the balloon, suggesting its involvement in estimating the risk of potential losses. Since the explosion of the balloon can be regarded as a change in the state of the balloon, as the balloon expands, the possibility of such a change in state (regime shift) also increases. In this view, the vmPFC result from *Schonberg et al., 2012* was consistent with our finding in that vmPFC negatively correlated with probability estimates of regime shift. Together, these results add to the existing literature by suggesting that vmPFC is involved in estimating and updating the state of the world in dynamic environments where changes take place regularly.

Related to OFC function in decision making and reinforcement learning, *Wilson et al., 2014* proposed that OFC is involved in inferring the current state of the environment. For example, medial OFC had been shown to represent probability distribution on possible states of the environment (*Chan et al., 2016*), the current task state (*Schuck et al., 2016*), and uncertainty or entropy associated with the state of the environment (*Muller et al., 2019*). In the context of regime-shift detection, regimes can be regarded as states of the environment and therefore a change in regime indicates a change in the state of the environment. *Muller et al., 2019* found that in dynamic environments where changes in the state of the environment happen regularly, medial OFC represented the level of uncertainty in the current state of the environment. Our finding that vmPFC represented individual participants' probability estimates of regime shifts suggests that vmPFC and/or OFC are involved in inferring the current state of the environment through estimating whether the state has changed. Our finding that vmPFC represented individual participants' sensitivity to transition probability further suggests that vmPFC and/or OFC contribute to individual participants' biases in state inference (over- and underreactions to change) in how these brain areas respond to the volatility of the environment.

Our results are also closely related to the literature on the neural mechanisms for evidence accumulation in decision making (*Gold and Shadlen, 2007*; *Mante et al., 2013*; *Philiastides et al., 2010*;

*Roitman and Shadlen, 2002*; *Yates et al., 2017*). In our task, evaluating the signals (red or blue balls) and, in particular, the strength of change evidence associated with the signals is central to performing the task. Normatively, such evaluation should depend on the signal diagnosticity. In a highly diagnostic environment, seeing a red ball should signal a strong possibility of being in the red regime, while seeing a blue ball should signal otherwise. By contrast, in a low diagnostic environment, a red (resp. blue) ball is not strongly indicative of a red (resp. blue) regime. Hence, the evaluation of signals should reflect the interaction between the signals and the diagnosticity of the signals.

We found that this key computation was implemented in a frontoparietal network commonly referred to as the frontoparietal control network (*Buckner et al., 2013*; *Dosenbach et al., 2007*; *Seeley et al., 2007*; *Vincent et al., 2008*; *Yeo et al., 2011*). This network was proposed to support adaptive control functions, including initiating control and providing flexibility to adjust the level of control through feedback (*Dosenbach et al., 2007*). The IPS and dlPFC, part of this network, have also been found to play a major role in the top-down control of attention (*Corbetta and Shulman, 2002*; *Woldorff et al., 2004*). In perceptual decision making, the IPS and dlPFC were also shown to represent the accumulation of sensory evidence that leads to the formation of a perceptual decision (*Heekeren et al., 2004*; *Heekeren et al., 2006*). Our findings—that activity in this network does not reflect just the sensory signals (red or blue balls) but how these signals should be interpreted through the lens of their diagnosticity—highlights the involvement of the frontoparietal control network in computing the strength of evidence through combining information about signals and knowledge about the precision of those signals.

For the frontoparietal network, we identified its involvement in our task through finding that its activity correlated with the strength of change evidence (*Figure 7*) and individual subjects' sensitivity to signal diagnosticity (*Figure 8*). Conceptually, these two findings reflect how individuals interpret the signals (signals consistent or inconsistent with change) in light of signal diagnosticity. This is because (1) strength of change evidence is defined as signals (+1 for signal consistent with change, and -1 for signal inconsistent with change) multiplied by signal diagnosticity and (2) sensitivity to signal diagnosticity reflects how individuals subjectively evaluate signal diagnosticity. At the theoretical level, these two findings can be interpreted through our computational framework in that both the strength of change evidence and sensitivity to signal diagnosticity contribute to estimating the likelihood of change (*Equations 1; 2* in *Materials and methods*).

Our result on the intraparietal sulcus (IPS) being part of the brain network that represents diagnosticity-weighted sensory signals is consistent with previous studies showing that IPS is involved in accumulating sensory evidence over time (*Gold and Shadlen, 2007*). There are three interesting aspects of our data that add to the current literature on evidence accumulation. First, IPS representations for sensory evidence need not be in the space of actions. Unlike previous studies showing that IPS represents sensory evidence for potential motor actions, we found that IPS represents the strength of evidence in favor of or against regime shifts. This result points to a more general role of the IPS in estimating the strength of sensory evidence. In fact, our result suggests that it depends on the task goal, which in the current study is to estimate whether a change has taken place. Second, although evidence accumulation is important and necessary for a wide array of cognitive functions, it is not a central requirement for the regime-shift task. Bayesian updating—the framework in which our system-neglect model was built upon—only requires the computation of the strength of change evidence associated with the signal shown in the latest period. By showing that IPS represents this quantity, this suggests that IPS is involved in evaluating the latest piece of evidence necessary for belief updating.

In the current study, the central opercular cortex—in addition to the vmPFC—is another brain region that represented the probability estimates of change. Like the vmPFC, activity in this region negatively correlated with the probability estimates of change. This finding is associated with previous findings on change detection using the oddball paradigm. Using the oddball paradigm, it was found that the central opercular cortex is involved in the detection of change, showing stronger activation in blocks containing only the standard stimulus than blocks containing both the standard and deviant stimulus (*Hedge et al., 2015*) and correlating with ERP P3 signals at the trial-level that reflected differences between standard and deviant stimuli (*Warbrick et al., 2009*). There are two implications here. First, our findings suggest that the central opercular cortex is not only involved in the detection of change—as revealed by the oddball tasks—but also is involved in the estimation of change where there is uncertainty regarding whether the state of the world had changed. Second, the central

opercular cortex may be part of a common pathway for the detection of change across very different tasks such as the oddball paradigm and our regime-shift detection task.

In the current study, our psychometric-neurometric analysis focused on comparing behavioral sensitivity with neural sensitivity to the system parameters (transition probability and signal diagnosticity). We measured sensitivity by estimating the slope of behavioral data (behavioral slope) and neural data (neural slope) in response to the system parameters. Previous studies had adopted a similar approach (*Ting et al., 2015*; *Vilares et al., 2012*; *Yang and Wu, 2020*). For example, *Vilares et al., 2012* found that sensitivity to prior information (uncertainty in prior distribution) in the orbitofrontal cortex (OFC) and putamen correlated with behavioral measures of sensitivity to the prior. In the current study, transition probability acts as prior in the system-neglect framework (*Equation 2* in *Materials and methods*), and we found that the ventromedial prefrontal cortex represents subjects' sensitivity to transition probability. Together, these results suggest that OFC (with vmPFC being part of OFC, see *Wallis, 2012*) is involved in the subjective evaluation of prior information in both static (*Vilares et al., 2012*) and dynamic environments (current study). In addition, distinct from vmPFC in representing sensitivity to transition probability or prior, we found through the behavioral-neural slope comparison that the frontoparietal network represents how sensitive individual decision makers are to the diagnosticity of signals in revealing the true state (regime) of the environment. Interestingly, such sensitivity to signal diagnosticity was only present in the frontoparietal network when participants encountered change-consistent signals. However, while most brain areas within this network responded in this fashion, only the left IPS and left IFG showed a significant difference in coding individual participants' sensitivity to signal diagnosticity between change-consistent and change-inconsistent signals. Unlike the left IPS and left IFG, we observed in dmPFC a marginally significant correlation with behavioral sensitivity at change-inconsistent signals as well. Together, these results indicate that while different brain areas in the frontoparietal network responded similarly to change-consistent signals, there was a greater degree of heterogeneity in responding to change-inconsistent signals.

In summary, our results suggest that an important mechanism for under- and overreactions to change has to do with neural sensitivity to system parameters that impact regime shifts. Importantly, different system parameters appear to recruit distinct brain networks according to their unique computational specializations. Given that under- and overreactions underlie a wide array of human judgments, our findings indicate that network-level computational specificity and parameter selectivity are two key building blocks that give rise to human judgment biases.

## Materials and methods

The data and analysis code are available at https://osf.io/xh7dy/.

We performed three fMRI experiments (90 subjects in total, 30 subjects for each experiment). Experiment 1 was the main experiment where we investigated the neurocomputational substrates for regime shifts. Experiments 2 and 3 were control experiments. Experiment 2 was designed to rule out brain activity that correlated with probability estimates but was not specifically about regime shifts. Experiment 3 attempted to rule out brain activity that correlated with entering numbers through button presses. In the main text, we only presented the results from Experiment 1. The procedure and results of Experiments 2 and 3 were presented in Supplementary Materials.

### Subjects

All subjects gave informed written consent to participate in the study. All subjects were right-handed. The study procedures were approved by the National Yang Ming Chiao Tung University Institutional Review Board (YM107054E). Ninety subjects participated in this study:

- Experiment 1: $n$=30 subjects; 15 males; mean age: 22.9 years; age range: 20–29 yrs.
- Experiment 2: $n$=30 subjects; 15 males; mean age: 23.3 years; age range, 20–30 years.
- Experiment 3: $n$=30 subjects; 15 males; mean age: 23.7 years; age range: 20–34 years.

Subjects were paid 300 New Taiwan dollars (TWD, 1 USD = 30 TWD) for participating in the behavioral session and 500 TWD for the fMRI session. Subjects received an additional monetary bonus based on his or her performance on probability estimation in Experiments 1 and 2 (Experiment 1: an average of 209 and 212 TWD for the behavioral and fMRI sessions, respectively; Experiment 2: an average of 223 and 206 TWD for the behavioral and fMRI sessions, respectively). In Experiment 3,

subjects received the bonus based on their performance for entering the correct number (an average of 243 TWD for the fMRI session).

## Procedure

### Overview

Experiment 1 consisted of two sessions—a behavioral session followed by an fMRI session—that took place on two consecutive days. Subjects performed the same task in both sessions. The goals of having the behavioral session were to familiarize subjects with the task and to have enough trials—along with the fMRI session—to reliably estimate the parameters of the system-neglect model. Details of Experiments 2 and 3 can be found in the Supplement.

### Regime-shift detection task

In this task, the environments the subjects are described as *regimes*. There were two possible regimes and at any point in time the regime can shift from one to another. Subjects judged whether the regime had shifted based on three sources of information: transition probability, signal diagnosticity, and signals. Prior to each trial, the subjects were given information about the transition probability and signal diagnosticity of the upcoming trial. The transition probability described how likely it is that the regime can shift from one to another. The signal diagnosticity described how different the two regimes are. The two regimes—red and blue—were represented by a red urn and a blue urn, respectively, with each urn consisting of red and blue balls. The red urn had more red balls and the blue urn had more blue balls. The two urns always shared the same ratio of the dominant ball color to the dominated ball color. That is, the ratio of the red balls to the blue balls in the red urn was the same as the ratio of the blue balls to the red balls in the blue urn. Signal diagnosticity was quantitatively defined by this ratio. For example, when the ratio was 9:1, signal diagnosticity ($d$) was 9. Under this definition, larger signal diagnosticity indicated that the two regimes were more different from each other.

After information about the transition probability and signal diagnosticity were revealed, 10 sensory signals—red or blue balls shown on the computer screen—were sequentially presented to the subjects. Each signal was sampled with replacement from one of the two urns. We also referred to the sequential presentation of these signals as *periods*. By design, each trial always started at the red regime, but the regime can shift from the red to the blue regime at any point in time during a trial, including prior to the sampling of the first signal. In addition, also by design, the regime can shift only once in a trial (i.e. the blue regime was an absorbing or trapping state). That is, once the regime changed from the red to the blue regime, signals would be sampled from the blue urn until the end of the trial. Subjects provided an estimate of the probability that the current regime was the blue regime at each period when a new signal was presented. Since the regime can only shift from the red to the blue, this probability estimate was equivalent to the probability estimate of regime shift.

### Stimuli

For each subject and each trial separately, we generated the stimuli, that is the sequence of red and blue balls according to the transition probability and signal diagnosticity of the trial. Before the start of each period, we first determined whether the regime would shift from the Red regime (the starting regime) to the Blue regime by sampling from the transition probability. There were two possible outcomes: 1 indicates a change in regime, whereas 0 indicates no change. If the outcome were 1, we would sample from the Blue regime for that period and for all the remaining period(s). If the outcome were 0, we would sample from the Red regime for that period and repeat the same process described above for the next period.

### Manipulations of transition probability and signal diagnosticity

We implemented a 3×3 factorial design where there were three possible transition probabilities, denoted as $q$ where $q = [0.01, 0.5, 0.1]$ and three levels of signal diagnosticity, denoted as $d$ where $d = [1.5, 3, 9]$. The transition probability—the probability that the regime shifts from the red to the blue regime—was used to simulate the stability of the environment. Larger transition probabilities indicate less stable environments. The signal diagnosticity was the ratio of dominant balls to the dominated balls in the urns. For example, when $d=9$, it indicates that the red balls are 9 times more than the blue

balls in the red urn and that the blue balls are 9 times more than the red balls in the blue urn. In this case, the two regimes were highly different and hence the signals shown to the subjects should be highly diagnostic of the regime the subjects were currently in.

## Session 1: behavioral session

A pre-test block followed by the main task (eight blocks of trials) was implemented in the behavioral session. We implemented the pre-test block to make sure the subjects understood the experimental instructions and to train them to become familiar with the task, especially the time limit in entering probability estimates (see Trial sequence below for details on the pre-test block). A 3 (transition probability) by 3 (signal diagnosticity) factorial design was implemented, resulting in a total of nine experimental conditions. Each block consisted of nine trials (one for each condition, randomized in order), and each trial consisted of 10 periods or equivalently, 10 sequentially presented signals. After the pre-test block, subjects performed the main task that consisted of eight blocks of trials. The behavioral session took approximately 70 min to complete.

### Trial sequence

At the beginning of each trial, the subjects were shown information about the transition probability and signal diagnosticity (3 s; *Figure 1A*). This was followed by the sequential presentation of 10 signals. At each period, a signal—a red or blue dot sampled from the current regime—was shown. Subjects' task was to estimate the probability that the current regime was the blue regime within 4 s. The subjects indicated the probability estimate through two button presses. During the experiment, the subjects placed his or her 10 fingers on a 10-button keypad. Each finger was associated with a single button that corresponded to a unique integer value from 0 to 9, with the left little finger for 1, left ring finger for 2, left middle finger for 3, left index finger for 4, left thumb for 5, right thumb for 6, right index finger for 7, right middle finger for 8, right ring finger for 9, and right little finger for 0. To enter the probability estimate, the subjects had to first press the number corresponding to the tens and second the number corresponding to the ones. For example, if the probability estimate was 95%, the subjects first had to press the '9' button and second the '5' button. Once the subjects entered the probability estimate, she or he was not allowed to change it. After providing the probability estimate, they were given brief feedback (0.5 s) on the number they just entered. If they failed to indicate the probability estimate within the time limit (4 s), a 'too slow' message would be shown. At the end of each trial, the subjects received feedback (2 s) on the amount of monetary bonus earned in the trial (2 s) and information about whether the regime shifted during the trial. If the regime was shifted, the signal that was drawn right after the shift took place would be highlighted in white. This was followed by a variable inter-trial interval (ITI, 1 s to 5 s in steps of 1 s drawn from a discrete uniform distribution).

We implemented a pre-test block at the beginning of the behavioral session to train the subjects to enter the probability estimate within the time limit (4 s). In the pre-test block, we started with a lenient time limit and gradually decreased it. The time limit for the first three trials was 20 s, 10 s from trial number 4–6, and 4 s from trial number 7–9. Recall that in each trial, the subjects encountered 10 sequentially presented signals (red and blue balls) and had to provide a probability estimate at each period. The subjects therefore entered 30 probability estimates under each of these three time limits—from 20 s, 10 s, to 4 s. After the pre-test block, all the subjects were able to enter the probability estimates within 4 s.

## Session 2: fMRI session

The fMRI session consisted of three blocks (nine trials in each block, with each trial consisting of 10 sequentially presented signals). The task was identical to the behavioral session (except with varying inter-stimulus intervals for the purpose of fMRI analysis) and took approximately 60 min to complete. The subjects indicated the probability estimate through two button presses. During the experiment, the subjects placed his or her 10 fingers on a 10-button keypad. Each finger was associated with a single button that corresponded to a unique integer value from 0 to 9 (starting from the left pinkie for 1, left ring finger for 2, left middle finger for 3, and etc., to right ring finger for 9, and finally right pinkie for 0). The trial sequence was identical to the behavioral session with a few exceptions. First, each new signal was presented for 4 s regardless of when subjects made a response. Second, we added a variable inter-stimulus interval (ISI, 1 s to 5 s in steps of 1 drawn from a uniform distribution)

between two successive signals. We also made the range of ITI to be slightly larger (1 s to 7 s drawn from a discrete uniform distribution in steps of 1 s) than the behavioral session. The design of variable ISIs and ITIs was to allow better dissociations between events, that is between different sensory signals presented during a trial, and between trials for fMRI analysis.

## Monetary bonus

To incentivize subjects to perform well in this experiment, they received a monetary bonus based on his or her probability estimates. The bonus rule was designed so that the subjects who gave more accurate estimates would earn more bonus. The bonus structure used a quadratic payoff:

$$Bonus\,(t) = \$30 \times \left(0.1 - 0.2 \times \left(P_t - B_t\right)^2\right),$$

where $P_t$ is the probability estimate that the current regime was blue at the $t$-th period in a trial and $B_t$ is the regime at $t$ ($B_t$=1 for the blue regime and $B_t$=0 for the red regime). For each probability estimate, the bonus therefore ranged from winning \$3 to losing \$3 TWD. For example, if the subject gave a 99% probability estimate that the current regime was blue and the current regime was indeed the blue regime, she would receive a bonus close to 3 TWD. By contrast, if the subjects gave a 1% probability estimate that the current regime was blue but the current regime was the red regime, she would receive a penalty close to 3 TWD. With 10 probability estimates given in each trial, the subjects can therefore receive a bonus of up to 30 TWD or a penalty of up to 30 TWD in a trial. The subjects did not receive feedback on the bonus after each probability estimate. Instead, at the end of each trial, the subjects received information about the total amount won or lost in that trial. The final total bonus was realized by randomly selecting 10 trials at the end of the experiment.

## Computational models for regime shift

We examined two computational models for regime shift: the Bayesian model and the system-neglect model (*Massey and Wu, 2005*). The Bayesian model was parameter-free and was used to compare with subjects' probability estimates of regime shift. The system-neglect model is a quasi-Bayesian model or parameterized version of the Bayesian model that was fit to the subjects' probability-estimate data. The parameter estimates of the system-neglect model were further used in the fMRI analysis so as to identify neural representations for over- and underreactions to change.

## Bayesian model

Here we describe the Bayesian posterior odds of shifting to the blue regime given the period history $H_t$ (*Edwards, 1968*; *Massey and Wu, 2005*):

$$\frac{P_t^B}{1 - P_t^B} = \frac{\Pr\left(B_t|H_t\right)}{\Pr\left(R|H_t\right)} = \frac{1 - \left(1-q\right)^t}{\left(1-q\right)^t} \sum_{j=1}^{t} \frac{q\left(1-q\right)^{j-1}}{1 - \left(1-q\right)^t} d^{t+1-j-2\sum_{k=j}^{t} r_k}, \tag{1}$$

$P_t^B$=$\Pr\left(B_t|H_t\right)$ is the posterior probability that the regime has shifted to the blue regime at the $t$-th period, $H_t$ denotes the sequence of history from $r_1$ to $r_t$. Here $r_t$ denotes the $t$-th period, where $r_t$=1 when the signal at $t$ is red, and $r_t$=0 when the signal at $t$ is blue. The transition probability and signal diagnosticity are denoted by $q$ and $d$, respectively.

The posterior odds are the product of the prior odds and the likelihood ratio. The prior odds, $\frac{1-\left(1-q\right)^t}{\left(1-q\right)^t}$, indicate that given the transition probability $q$, the probability that the regime has shifted at time $t$, $1 - \left(1-q\right)^t$, relative to the probability of no change in regime, $\left(1-q\right)^t$. The likelihood ratio, $\sum_{j=1}^{t} \frac{q\left(1-q\right)^{j-1}}{1-\left(1-q\right)^t} d^{t+1-j-2\sum_{k=j}^{t} r_k}$, indicates the probability of observing the history of signals $H_t$ given that the regime has shifted relative to the probability given that the regime has not shifted. This requires considering all the possibilities on the timing of the shift, that is the likelihood ratio that the regime was shifted at $t$=$j$, $d^{t+1-j-2\sum_{k=j}^{t} r_k}$, weighted by its odds $\frac{q\left(1-q\right)^{j-1}}{1-\left(1-q\right)^t}$. Since these possibilities are disjoint

events, the likelihood ratio that the regime has shifted is simply the weighted sum of all the likelihood ratios associated with these disjoint possibilities.

## Index of overreaction to change

To quantify under- and overreactions to change, we derived an Index of Overreaction (*IO*). In short, this index reflects the degree to which the subjects overreacted or underreacted to change such that an index value greater than 0 indicates overreaction and an index value smaller than 0 suggests underreaction. To compute *IO*, we compared the change in probability estimates between two adjacent periods ($\Delta P_t = P_t - P_{t-1}$) with the normative change in probability estimates according to the Bayesian model ($\Delta P_t^B = P_t^B - P_{t-1}^B$). Here we use $P_t$ to denote the subject's probability estimate at the $t$-th period and $P_t^B$ to denote the Bayesian probability estimate. We computed *IO* ($IO = \Delta P_t - \Delta P_t^B$) separately for each subject and for each condition or combination of transition probability and signal diagnosticity. Overreaction is defined when the actual change in probability estimates is greater than the normative change in probability estimates ($\Delta P_t > \Delta P_t^B$). By contrast, underreaction is defined when the actual change is smaller than the normative change in probability estimate ($\Delta P_t < \Delta P_t^B$). When $\Delta P_t^e = \Delta P_t^B$, there is neither overreaction nor underreaction to change.

## System-neglect model

Following *Massey and Wu, 2005*, we fit the system-neglect model—a quasi-Bayesian model—to the subjects' probability estimates. In this model, we aimed to capture system-neglect—that people respond primarily to the signals and secondarily to the system generating the signals. Responding secondarily to the system indicates that people with system-neglect would be less sensitive to the changes in the system parameters compared with the normative Bayesian. This is captured by adding a weighting parameter to each system parameter level. Hence, two weighting parameters, $\alpha$ and $\beta$, were added to *Equation (1)* to transition probability and signal diagnosticity respectively such that

$$\frac{P_t}{1 - P_t} = \frac{\Pr\left(B_t|H_t\right)}{\Pr\left(R|H_t\right)} = \frac{1 - \left(1 - \alpha q\right)^t}{\left(1 - \alpha q\right)^t} \sum_{j=1}^{t} \frac{q\left(1 - q\right)^{j-1}}{1 - \left(1 - q\right)^t} d^{\beta\left(t+1-j-2\sum_{k=j}^{t} r_k\right)} \tag{2}$$

where $P_t$ is the probability estimate that regime has changed to the blue regime at period $t$. We separately estimated $\alpha$ for each level of transition probability and $\beta$ for each level of signal diagnosticity. This was implemented by setting dummy variables for each level of transition probability and signal diagnosticity in *Equation 2*.

$$\alpha = \alpha_1 Q_1 + \alpha_2 Q_2 + \alpha_3 Q_3 \text{ and } \beta = \beta_1 D_1 + \beta_2 D_2 + \beta_3 D_3.$$

where $Q_i$ is the dummy variable for transition probability $q_i$ and $D_j$ is the dummy variable for diagnosticity $d_j$. $\alpha_i$ and $\beta_j$ therefore respectively reflect the sensitivity to different levels of transition probability and signal diagnosticity. For each subject separately, we performed nonlinear regression (using the *fitnlm* function in MATLAB) to fit the model to the subject's probability estimates and estimated the parameters of interest.

## Parameter recovery analysis

To examine whether the fitting procedure gave reasonable parameter estimates of the system-neglect model, we performed a parameter recovery analysis (*Wilson and Collins, 2019*). The analysis proceeded in the following steps. First, we simulated each subject's probability estimation data based on the system-neglect model by using the parameter estimates obtained for that subject after fitting the system-neglect model to the subject' probability estimation data. Second, we fit the system-neglect model to the simulated data. Third, as a measure of parameter recovery, we computed the correlation across subjects between the estimated parameters and the parameter values we used to simulate data, where larger correlation indicates better recovery. Fourth, we repeated the above steps by adding independent white noise to the simulated data, from a Gaussian distribution with mean 0 and variance $\sigma^2$. We implemented five levels of noise with $\sigma_{noise} = \{0.01, 0.05, 0.1, 0.2, 0.3\}$ and examined the impact of noise on parameter recovery. These noise levels covered the range of empirical noise levels estimated from the subjects. To estimate each subject's noise level, we incorporated a noise parameter into the system-neglect model. We assumed that probability estimates are noisy and

modeled them with a Gaussian distribution where the noise parameter $(\sigma_{noise})$ is the standard deviation. At each period, a probability estimate of regime shift was computed according to the system-neglect model where $\Theta$ is the set of parameters including parameters in the system-neglect model and the noise parameter. The likelihood function, $L(\Theta)$, is the probability of observing the subject's probability estimate at period $t$, $p_t$, given $\Theta$, $L(\Theta) = P(p_t|\Theta)$. Since we modeled the noisy probability estimates with a Gaussian distribution, we can therefore express $L(\Theta)$ as $L(\Theta) \sim N\left(p_t; p_t^{SN}, \sigma_{noise}\right)$ where $p_t^{SN}$ is the probability estimate predicted by the system-neglect (SN) model at period $t$. As a reminder, we referred to a 'period' as the time when a new signal appeared during a trial (for a given transition probability and signal diagnosticity). To find the maximum likelihood estimates of the parameters, $\Theta_{\mathrm{MLE}}$, we summed over all periods the negative natural logarithm of likelihood and used MATLAB's fmincon function to find $\Theta_{\mathrm{MLE}}$. Across subjects, we found that the mean noise estimate was 0.1735 and ranged from 0.1118 to 0.2704 (*Figure 3*).

## Impact of noise homoscedasticity on parameter estimation

The fitnlm function we used in MATLAB to estimate the parameters of the system-neglect models assumes homoscedasticity—that noise is constant when fitting the model. However, it is possible that this assumption was violated, that subjects' actual noise was heteroscedastic, for example, having larger noise when probability estimates are around 0.5 and smaller noise at the two extremes (0 and 1). To examine this possibility, for each subject, we divided probability into five intervals ([0.0–0.2), [0.2–0.4), [0.4–0.6), [0.6–0.8), and [0.8–1.0]) and computed the residual standard deviation separately for each bin. Here, the residual was the difference between subjects' probability estimates $(p_t)$ and the probability estimates derived from the system-neglect model based on each subject's parameter estimates $\left(p_t^{SN}\right)$, $residual = p_t - p_t^{SN}$. We found that homoscedasticity was indeed violated—the standard deviation of residuals was smallest when probability was 0.1 and increased as a function of probability from 0.1 to 0.5. When probability was larger than 0.5, the residual standard deviation was similar (*Figure 3G*). To see how this would affect parameter estimation, we performed parameter recovery analysis assuming heteroscedasticity. That is, we simulated subjects' probability estimates using the empirically estimated, probability-dependent residual standard deviation as the standard deviation of the Gaussian noise and using the simulated data to estimate parameters in the system-neglect model. We found that we were able to recover the parameters well (*Figure 3H*) and the result was similar to the parameter recovery assuming homoscedastic noise (*Figure 3A–E*). This suggested that heteroscedastic noise did not impact the accuracy of parameter estimation when homoscedasticity was assumed.

### fMRI data acquisition

All the imaging parameters, including the EPI sequence and MPRAGE sequence, remained the same throughout the three experiments. For Experiments 1 and 2, the subjects completed the task in a 3T Siemens MRI scanner (MAGNETOM Trio) equipped with a 32-channel head array coil. Experiment 3 was collected at a later date after the same scanner went through a major upgrade (from Trio to the Prisma system) where a 64-channel head array coil was used. Each subject completed three functional runs. Before each run, a localizer scan was implemented for slice positioning. For each run, T2*-weighted functional images were collected using an EPI sequence (TR = 2000 ms, TE = 30 ms, 33 oblique slices acquired in ascending interleaved order, 3.4×3.4 × 3.4 mm isotropic voxel, 64×64 matrix in 220 mm field of view, flip angle 90°). To reduce signal loss in the ventromedial prefrontal cortex and orbitofrontal cortex, the sagittal axis was tilted clockwise up to 30°. Each run consisted of 9 trials and a total of 374 images. After the functional scans, T1-weighted structural images were collected (MPRAGE sequence with TR = 2530ms, TE = 3.03 ms, flip angle = 7°, 192 sagittal slices, 1 × 1 × 1 mm isotropic voxel, 224×256 matrix in a 256 mm field of view). For each subject, a field map image was also acquired for the purpose of estimating and partially compensating for geometric distortion of the EPI image so as to improve registration performance with the T1-weighted images.

### fMRI preprocessing

The imaging data were preprocessed with FMRIB's Software Library (FSL version 6.0). First, for motion correction, MCFLIRT was used to remove the effect of head motion during each run. Second, FUGUE

(FMRIB's Utility for Geometric Unwarping of EPIs) was used to estimate and partially compensate for geometric distortion of the EPI images using field map images collected for the subject. Third, spatial smoothing was applied with a Gaussian kernel with FWHM = 6 mm. Fourth, a high-pass temporal filtering was applied using Gaussian-weighted least square straight-line fitting with $\sigma = 50s$. Fifth, registration was performed in a two-step procedure, with the field map used to improve the performance of registration. First, EPI images were registered to the high-resolution brain T1-weighted structural image; non-brain structures were removed via FSL's BET (Brain Extraction Tool). Second, the transformation matrix (12-parameter affine transformation) from the T1-weighted image to the Montreal Neurological Institute (MNI) template brain was estimated using FLIRT (FMRIB's Linear Image Registration Tool), followed by nonlinear registration using FNIRT (FMRIB's Non-linear Image Registration Tool) with a 10 mm warp resolution. This two-step procedure allowed for transforming the EPI images to the standard MNI template brain.

## General Linear Models of BOLD signals

All GLM analyses were carried out in the following steps (*Beckmann et al., 2003*). First, BOLD time series were pre-whitened with local autocorrelation correction. A first-level FEAT analysis was carried out for each run of each subject. Second, a second-level (subject-level) fixed-effect (FE) analysis was carried out for each subject that combined the first-level FEAT results from different runs using the summary statistics approach. Finally, a third-level (group-level) mixed-effect (ME) analysis using FSL's FLAME module (FMRIB's Local Analysis of Mixed Effects) was carried out across subjects by taking the FE results from the previous level and treating subjects as a random effect (*Woolrich et al., 2004*). All reported whole-brain results were corrected for multiple comparisons. We first identified clusters of activation by defining a cluster-forming threshold of the z statistic ($z$>3.1 or equivalently $p$ < 0.001; *Eklund et al., 2016*; *Woo et al., 2014*). Then, a family-wise error corrected p-value of each cluster based on its size was estimated using Gaussian random field theory (*Worsley et al., 1992*). In addition, we performed a nonparametric permutation test using the randomize function in FSL (threshold-free cluster enhancement or TFCE option *Smith and Nichols, 2009*) on all the contrasts reported.

### GLM-1

This model was used for *Figures 5 and 6*. The model served two purposes. First, we used it to examine neural representations for probability estimates ($P_t$) and belief revision ($\Delta P_t$)(*Figure 5AB*). Second, we used this model to compare results between the main experiment (Experiment 1) and the control experiments (Experiments 2 and 3; *Figure 5CD*). We implemented the following regressors. At the time of each signal presentation, we implemented the following regressors: (R1) an indicator regressor with length equal to the subject's RT, (R2) R1 multiplied by the subject's probability estimate, $P_t$, that the signal came from the Blue regime, (R3) R1 multiplied by the difference in the subject's probability estimate between two successive periods, $\Delta P_t$, which captured the updating of belief about change, (R4) R1 multiplied by the degree of certainty in probability estimate l$P_t$ − 0.5l, (R5) R1 multiplied by the period number (from 1 to 10). Both positive and negative contrasts of R2 ($P_t$) and R3 ($\Delta P_t$) were set up to identify activity that either positively or negatively correlates with these regressors. At the end of each trial (after subjects saw all 10 signals), we provided the monetary bonus the subject earned in that trial. We implemented an indicator regressor (R6) and a parametric regressor for the subject's winning (R7). We implemented an indicator regressor (length equal to 4 s) for the no-response periods (R8), which corresponded to the period(s) where the subject did not indicate the probability estimate within the time limit (4 s). Finally, to directly address the motor confound issue, we implemented an action-handedness regressor, (R9), which was R1 multiplied by action-handedness. This regressor served to address the motor confounds of probability estimates $P_t$, as higher probability estimates preferentially involved right-handed responses for entering higher digits and lower estimates involved left-handed responses. Therefore, at the time of each signal presentation, the action-handedness regressor coded –1 if both finger presses to enter $P_t$ involved the left hand, 0 if using one left finger and one right finger, and 1 if both finger presses involved the right hand. For Experiment 1, we also performed a GLM analysis that was identical to GLM-1 except that it did not include the action-handedness regressor and the results on both $P_t$ and $\Delta P_t$ were largely identical (see *Supplementary file 12*). Note that for both Experiment 1 (the main regime-shift experiment) and Experiment 2 (control experiment), $P_t$ indicates the probability estimate that the signal came from

the Blue regime. However, for Experiment 3 (control experiment), subjects were instructed to press a two-digit number shown on the screen at each period. Hence, the $P_t$ regressor for Experiment 3 was effectively the instructed number, and the $\Delta P_t$ regressor was effectively the difference in instructed number between adjacent periods.

## GLM-2

This model was implemented to examine the effects of component variables critically contributing to regime-shift probability estimation. In particular, we used this model to identify the effects on the strength of change evidence (based on R11 below) and the intertemporal prior (based on R9 below) (*Figure 7*). We also used this model to compare the effect of $P_t$ and $\Delta P_t$ with GLM-1 to examine the robustness of $P_t$ and $\Delta P_t$ representations in vmPFC and ventral striatum (*Figure 6*). The model was identical to GLM-1 from (R1) to (R8) with the addition of the following regressors: (R9) R1 multiplied by the intertemporal prior, defined as $\ln\left(\frac{1-(1-q)^t}{(1-q)^t}\right)$, where $q$ is the transition probability and $t$ is the period, (R10) R1 multiplied by the natural logarithm of signal diagnosticity, $\ln(d)$, (R11) R1 multiplied by the current signal (1 for blue ball, –1 for red ball), and (R12) the interaction $R9 \times R10$ (interaction between signal diagnosticity and signal). We note that since these component variables (R9 to R12) contributed to regime-shift probability estimation, they were correlated with $P_t$ and $\Delta P_t$. Having both with $P_t$ and $\Delta P_t$ in the same model as these component variables would therefore introduce collinearity and reduce the reliability of regression coefficients. While we were aware of this issue, we also recognized that having these correlated regressors in the same model has the advantage of statistically stating that neural activity attributed to a particular variable cannot be otherwise attributed to another variable that correlated with it. We also performed another GLM analysis that was identical to GLM-2 except that it did not include $P_t$ and $\Delta P_t$ in the model. The results on strength of change evidence and intertemporal prior were largely identical to those shown in *Figure 7*. See *Supplementary file 13* for results of this model.

## GLM-3

This model was the basis of results shown in *Figure 8*. We set up two sets of regressors, one set for when the blue signal (signal for potential change) appeared and the other for when the red signal (signal for no change) appeared. For each set, at the time of signal presentation, we included nine indicator regressors, one for every combination of three transition probability levels and three signal diagnosticity levels. Based on these nine regressors, we set up a linear-increasing contrast separately for signal diagnosticity and transition probability. At the subject level (second level), the parameter estimate of these contrasts reflects individual subjects' sensitivity to signal diagnosticity and transition probability, which were extracted from a given ROI and were used to correlate with individual subjects' behavioral sensitivity to signal diagnosticity and transition probability. The ROIs used were vmPFC, ventral striatum, dmPFC, bilateral IFG, and bilateral IPS. These ROIs were identified by results from GLM 1 and 2 and were constructed in a statistically independent manner using the leave-one-subject-out method. See Independent region-of-interest (ROI) analysis in the section below. Same as in the previous two GLMs, at the end of each trial when feedback on the current-trial monetary bonus was revealed, we implemented an indicator regressor and a parametric regressor for the monetary bonus. Finally, we implemented an indicator regressor (length equal to 4 s) for no-response periods. These corresponded to the period(s) where the subject did not indicate the probability estimate within the time limit (4 s).

## Independent regions-of-interest (ROIs) analysis

We performed two kinds of independent ROI analysis—leave-one-subject-out (LOSO) method and functional/structural masks based on previous meta-analysis paper (*Bartra et al., 2013*) or existing structural atlases. Specifically, we used the functional mask in vmPFC from *Bartra et al., 2013* and structural mask for the ventral striatum based on the Harvard-Oxford cortical and subcortical atlases in FSL for analysis shown in *Figures 5, 6 and 8*. For LOSO, based on results from the whole-brain analysis, we created independent and unbiased ROIs using the leave-one-subject-out (LOSO) method (*Litt et al., 2011*; *Ting et al., 2015*). The LOSO method was used to analyze probability estimates and representations for transition probability and signal diagnosticity described above. We performed the

analysis for each subject separately in the following steps. First, we identified the significant cluster in a brain region (e.g. dmPFC) that correlated with a contrast of interest (e.g. probability estimates) using all other subjects' data. We referred to this as the LOSO ROI. Second, we extracted the mean beta value (regression coefficient) within the LOSO ROI from the subject and used it for further statistical analysis. Note that the LOSO ROI tended to be different spatially between subjects. To give an idea of these differences and the spatial distribution of these ROIs, in *Figures 7 and 8* where the LOSO analysis was performed, we showed both the voxels that were part of the LOSO ROI of all the subjects in one color and the voxels that were part of at least one LOSO ROI in another color. Finally, we note that since the LOSO procedure involves, for each subject separately, performing the leave-one-subject-out inference at the group level and identifying the significant clusters of activation. Therefore, it is possible that some subject(s), after the LOSO inference, did not have a significant cluster in some brain region. For the analysis shown in *Figures 7 and 8*, for the left IFG ROI, we had three subjects that did not have a significant cluster in this brain region. All other ROIs had data from all subjects.

## Parametric and nonparametric tests for difference in correlation coefficients

We implemented both parametric and nonparametric tests to examine whether the difference in Pearson correlation coefficients was significant. We denote the correlation coefficient between neural and behavioral sensitivity at change-consistent (blue) signals as $r_{blue}$, and that at change-inconsistent (red) signals as $r_{red}$. In the parametric test, we adopted the approach of *Meng et al., 1992* to statistically compare the two correlation coefficients. This approach specifically tests differences between dependent correlation coefficients according to the following equation

$$z = (z_{r1} - z_{r2}) \sqrt{\frac{N-3}{2(1-r_x)h}}$$

where $N$ is the number of subjects, $z_{ri}$ is the Fisher z-transformed value of $r_i$, ($r_1 = r_{blue}$ and $r_2 = r_{red}$), and $r_x$ is the correlation between the neural sensitivity at change-consistent signals and change-inconsistent signals. The computation of $h$ is based on the following equations

$$h = \frac{1 - f\overline{r^2}}{1 - \overline{r^2}} = 1 + \frac{\overline{r^2}}{1 - \overline{r^2}}(1-f)$$

$$f = \frac{1 - r_x}{2(1 - \overline{r^2})}, \; which \, must \, be \leq 1$$

where $\overline{r^2}$ is the mean of the $r_i^2$, $(r_1^2 + r_2^2)/2$, and $f$ should be set to 1 if >1. In the nonparametric test, we performed nonparametric bootstrapping to test for the difference in correlation (*Efron and Tibshirani, 1994*). That is, we resampled with replacement the dataset (subject-wise) and used the resampled dataset to compute the difference in correlation. We then repeated the above for 100,000 times so as to estimate the distribution of the difference in correlation coefficients, tested for significance, and estimated p-value based on this distribution.

# Acknowledgements

This work was supported by the National Science and Technology Council (NSTC) in Taiwan (Grants 108-2410-H-010-012-MY3, 110-2410-H-A49A-504 -MY3 to S-WW) and by the Brain Research Center, National Yang Ming Chiao Tung University from The Featured Areas Research Center Program within the framework of the Higher Education Sprout Project by the Ministry of Education (MOE) in Taiwan. We acknowledge magnetic resonance imaging support from National Yang Ming Chiao Tung University, Taiwan, which is in part supported by the Ministry of Education plan for the top University.

## Additional information

### Funding

| Funder | Grant reference number | Author |
|---|---|---|
| National Science and Technology Council | 108-2410-H-010-012-MY3 | Shih-Wei Wu |
| National Science and Technology Council | 110-2410-H-A49A-504 -MY3 | Shih-Wei Wu |

The funders had no role in study design, data collection and interpretation, or the decision to submit the work for publication.

### Author contributions

Mu-Chen Wang, Conceptualization, Data curation, Formal analysis, Validation, Investigation, Visualization, Methodology, Writing – original draft, Writing – review and editing; George Wu, Conceptualization, Formal analysis, Supervision, Visualization, Methodology, Writing – original draft, Writing – review and editing; Shih-Wei Wu, Conceptualization, Resources, Formal analysis, Supervision, Funding acquisition, Validation, Investigation, Visualization, Methodology, Writing – original draft, Writing – review and editing

### Author ORCIDs

Mu-Chen Wang ⓘ https://orcid.org/0009-0004-9451-2193
Shih-Wei Wu ⓘ https://orcid.org/0000-0002-2728-9620

### Ethics

All subjects gave informed written consent to participate in the study. All subjects were right-handed. The study procedures were approved by the National Yang Ming Chiao Tung University Institutional Review Board.

Reviewer #1 (Public review): https://doi.org/10.7554/eLife.104684.5.sa1
Reviewer #3 (Public review): https://doi.org/10.7554/eLife.104684.5.sa2
Author response https://doi.org/10.7554/eLife.104684.5.sa3

## Additional files

### Supplementary files

Supplementary file 1. Experiment 1: Probability estimates $(P_t)$ and belief revision $(\Delta P_t)$ contrasts based on GLM-1. Cluster-level inference using Gaussian random field theory (familywise error corrected at p < 0.05 with a cluster-forming threshold $z > 3.1$).

Supplementary file 2. Experiment 1: Probability estimates $(P_t)$ and belief revision $(\Delta P_t)$ contrasts based on GLM-1. Permutation tests based on threshold-free-cluster-enhancement (TFCE) statistic.

Supplementary file 3. Experiment 1: Probability estimates $(P_t)$ and belief revision $(\Delta P_t)$ contrasts based on GLM-1. Permutation tests based on cluster extent.

Supplementary file 4. Experiment 3: Instructed number $(IN_t)$ and difference in instructed number $(\Delta IN_t)$ contrasts based on GLM-1. For Experiment 3, $IN_t$ represents the two-digit number subjects were instructed to press at each period, and $\Delta IN_t$ represents the difference in number between successive periods. $IN_t$ is the control for $P_t$ in Experiment 1, and $\Delta IN_t$ is the control for $\Delta P_t$ in Experiment 1. Cluster-level inference using Gaussian random field theory (familywise error corrected at p < 0.05 with a cluster-forming threshold $z > 3.1$).

Supplementary file 5. Experiment 1 and Experiment 2 based on the probability estimates $(P_t)$ contrast in GLM-1. Cluster-level inference using Gaussian random field theory (familywise error corrected at p < 0.05 with a cluster-forming threshold $z > 3.1$).

Supplementary file 6. Experiment 1 and Experiment 3: based on the probability estimates $(P_t)$ contrast in GLM-1. Cluster-level inference using Gaussian random field theory (familywise error corrected at p<0.05 with a cluster-forming threshold z>3.1).

Supplementary file 7. Experiment 1: GLM-2. Cluster-level inference using Gaussian random field theory (familywise error corrected at p < 0.05 with a cluster-forming threshold $z>3.1$).

Supplementary file 8. Experiment 1: GLM-2. Permutation tests based on the threshold-free-cluster-enhancement (TFCE) statistic.

Supplementary file 9. Experiment 1: GLM-2. Permutation tests based on cluster extent.

Supplementary file 10. Model-fitting summary.

Supplementary file 11. Model comparison using paired t-test with Bonferroni correction.

Supplementary file 12. Experiment 1: GLM-1 without the action-handedness regressor. Cluster-level inference using Gaussian random field theory (familywise error corrected at $p<.05$ with a cluster-forming threshold $z>3.1$).

Supplementary file 13. Experiment 1: GLM-2 without $P_t$ and $\Delta P_t$ regressors. Cluster-level inference using Gaussian random field theory (familywise error corrected at p < 0.05 with a cluster-forming threshold $z>3.1$).

MDAR checklist

## Data availability

All data, including behavioral and fMRI, and analysis code are available at Open Science Framework: https://osf.io/xh7dy/.

The following dataset was generated:

| Author(s) | Year | Dataset title | Dataset URL | Database and Identifier |
|---|---|---|---|---|
| Wang M-C, Wu G, S-W Wu | 2026 | System Neglect and the Neurocomputational Substrates for Over- and Underreactions to Changes | https://osf.io/xh7dy | Open Science Framework, xh7dy |

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

# Appendix 1

## Methods for Experiments 2 and 3
### Experiment 2
The procedure of Experiment 2 was identical to Experiment 1, except that there was no regime shift in this experiment. In other words, we only manipulated the signal diagnosticity in this experiment. The regime that the sensory signals were drawn from (red or blue urn) was randomly determined such that in half of the trials the regime was red and the other half the regime was blue. The order of the regimes across trials was pseudo-randomized for each subject separately. Identical to Experiment 1, the subjects in Experiment 2 were instructed to estimate the probability that the regime was the blue urn at the presentation of each new signal. Therefore, in both Experiments 1 and 2, the subjects estimated the probability that the current regime was the blue regime. However, it was only in Experiment 1 that the probability estimates conveyed information about the subjects' belief about change (whether the regime had shifted).

### Experiment 3
We referred to Experiment 3 as the motor-equivalent task of Experiments 1 and 2. The primary goal of this experiment was to rule out the motor confounds for probability estimates in Experiment 1. We were aware that brain regions whose activity correlated with probability estimates in Experiment 1 can simply reflect entering of numbers through button presses and thus have nothing to do with estimating the probability of change. Therefore, to establish the evidence for the neural representations of probability estimates, we needed to rule out the motor confounds. The motor-equivalent task shared many of the physical and motor aspects of the tasks in Experiments 1 and 2. First, the task structure was identical. In the motor-equivalent task, each trial contained ten rounds. In each round, a hollow dot with no color information was shown to indicate the number of this round. A random number from 0 to 99 was presented at the center of the screen. Subjects had to enter the indicated number with two successive button presses within 4 s. After this button-press stage, there was feedback on the number they just entered, with white given for correct inputs and yellow given for incorrect entries. The reward was $3 for each correct answer and -$3 for incorrect answers. The accumulated reward outcome for each trial was revealed at the end of the trial.

## Computational models for Experiment 2
Below we describe the computational models (Bayesian model and system-neglect model) for Experiment 2 where subjects performed identical tasks as in Experiment 1 with one exception that no regime shift was possible (transition probability $q$=0). We fit the system-neglect model, and the parameter estimates can be found in *Figure 2E*.

## Bayesian model
The Bayesian posterior odds of the blue regime at the $t$-th period given the signal history $H_t$ were calculated by:

$$\frac{P_t^B}{1 - P_t^B} = \frac{\Pr\left(B_t|H_t\right)}{\Pr\left(R|H_t\right)} = \frac{\Pr\left(B\right)}{\Pr\left(R\right)} \times \frac{\Pr\left(H_t|B\right)}{\Pr\left(H_t|R\right)} \tag{3}$$

where $\frac{\Pr(B)}{\Pr(R)}$ is the prior odds and $\frac{\Pr(H_t|B)}{\Pr(H_t|R)}$ is the likelihood ratio. Given that the probability of being in the blue regime and being in the red regime were equal $\Pr\left(B\right)$ =$\Pr\left(R\right)$ =0.5, *Equation (3)* can be rewritten as

$$\frac{P_t^B}{1 - P_t^B} = \frac{\Pr\left(B_t|H_t\right)}{\Pr\left(R|H_t\right)} = \frac{0.5}{0.5} \times \frac{\Pr\left(H_t|B\right)}{\Pr\left(H_t|R\right)} = \frac{\Pr\left(H_t|B\right)}{\Pr\left(H_t|R\right)}.$$

The likelihood ratio is computed according to the following equation

$$\frac{\Pr\left(H_t|B\right)}{\Pr\left(H_t|R\right)} = d^{t-2\sum_{k=1}^{t} r_k}.$$

where $d$ represents signal diagnosticity. For $r_k$, $r_k=1$ if subjects were presented with the red ball and $r_k=0$ if subjects were presented with the blue ball in the $k$-th period. The exponent of $d$ captures the difference between the number of the blue balls and red balls from the first to the $t$-th period. For example, suppose that at the fifth period there are four red balls and one blue ball. Then $d^{t-2\sum_{k=1}^{t} r_k}$ would be $d^{5-2\times4}=d^{-3}$ where the exponent is the number of blue balls minus the number of red balls.

## System-neglect model

Similar to Experiment 1, we developed and fit the system-neglect model. The system-neglect model consisted of a weighting parameter $\beta$ for signal diagnosticity

$$\frac{P_t^B}{1-P_t^B} = \frac{\Pr\left(B_t|H_t\right)}{\Pr\left(R|H_t\right)} = d^{\beta\left(t-2\sum_{k=1}^{t} r_k\right)}.$$

where we separately estimated $\beta$ for each level of signal diagnosticity

$$\beta = \beta_1 D_1 + \beta_2 D_2 + \beta_3 D_3$$

Here, $D_n$ is the dummy variable for signal diagnosticity $d_n$. Note that $\beta=1$ corresponds to the normative Bayesian $\beta$.

