## [Editor Report · eLife Assessment]

This study offers **valuable** insights into how humans detect and adapt to regime shifts, highlighting dissociable contributions of the frontoparietal network and ventromedial prefrontal cortex to sensitivity to signal diagnosticity and transition probabilities. The combination of an innovative instructed-probability task, Bayesian behavioral modeling, and model-based fMRI analyses provides **solid** support for the main claims. The addition of new model-comparison figures in revision effectively addresses the previously noted potential confound between posterior switch probability and time in the neuroimaging results. At the behavioral level, while the computational model captures the pattern of "system neglect" well, qualitatively distinct mechanisms, such as hyper-prior attraction toward experiment-wise mean parameters, reporting biases, or probability-outlier underweighting, could produce similar behavioral signatures and cannot be fully disambiguated with the current design alone; however, converging evidence from the authors' prior work partially mitigates this concern.

---

## [Referee Report · Reviewer #1 (Public review)]

Summary:

The study examines human biases in a regime-change task, in which participants have to report the probability of a regime change in the face of noisy data. The behavioral results indicate that humans display systematic biases, in particular, overreaction in stable but noisy environments, and underreaction in volatile settings with more certain signals. fMRI results suggest that a frontoparietal brain network is selectively involved in representing subjective sensitivity to noise, while the vmPFC selectively represents sensitivity to the rate of change.

Strengths:

- The study relies on a task that measures regime-change detection primarily based on descriptive information about the noisiness and rate of change. This distinguishes the study from prior work using reversal-learning or change-point tasks in which participants are required to learn these parameters from experiences. The authors discuss these differences comprehensively.

- The study uses a simple Bayes-optimal model combined with model fitting, which seems to describe the data well. The model is comprehensively validated.

- The authors apply model-based fMRI analyses that provide a close link to behavioral results, offering an elegant way to examine individual biases.

Weaknesses:

The authors have adequately addressed my prior concerns.

---

## [Referee Report · Reviewer #3 (Public review)]

This study concerns how observers (human participants) detect changes in the statistics of their environment, termed regime shifts. To make this concrete, a series of 10 balls are drawn from an urn that contains mainly red or mainly blue balls. If there is a regime shift, the urn is changed over (from mainly red to mainly blue) at some point in the 10 trials. Participants report their belief that there has been a regime shift as a % probability. Their judgment should (mathematically) depend on the prior probability of a regime shift (which is set at one of three levels) and the strength of evidence (also one of three levels, operationalized as the proportion of red balls in the mostly-blue urn and vice versa). Participants are directly instructed of the prior probability of regime shift and proportion of red balls, which are presented on-screen as numerical probabilities. The task therefore differs from most previous work on this question in that probabilities are instructed rather than learned by observation, and beliefs are reported as numerical probabilities rather than being inferred from participants' choice behaviour (as in many bandit tasks, such as Behrens 2007 Nature Neurosci).

The key behavioural finding is that participants over-estimate the prior probability of regime change when it is low, and under estimate it when it is high; and participants over-estimate the strength of evidence when it is low and under-estimate it when it is high. In other words participants make much less distinction between the different generative environments than an optimal observer would. This is termed 'system neglect'. A neuroeconomic-style mathematical model is presented and fit to data.

Functional MRI results how that strength of evidence for a regime shift (roughly, the surprise associated with a blue ball from an apparently red urn) is associated with activity in the frontal-parietal orienting network. Meanwhile at time-points where the probability of a regime shift is high, there is activity in another network including vmPFC. Both networks show individual differences effects, such that people who were more sensitive to strength of evidence and prior probability show more activity in the frontal-parietal and vmPFC-linked networks respectively.

Strengths

(1) The study provides a different task for looking at change-detection and how this depends on estimates of environmental volatility and sensory evidence strength, in which participants are directly and precisely informed of the environmental volatility and sensory evidence strength rather than inferring them through observation as in most previous studies

(2) Participants directly provide belief estimates as probabilities rather than experimenters inferring them from choice behaviour as in most previous studies

(3) The results are consistent with well-established findings that surprising sensory events activate the frontal-parietal orienting network whilst updating of beliefs about the word ('regime shift') activates vmPFC.

Weaknesses

(1) The use of numerical probabilities (both to describe the environments to participants, and for participants to report their beliefs) may be problematic because people are notoriously bad at interpreting probabilities presented in this way, and show poor ability to reason with this information (see Kahneman's classic work on probabilistic reasoning, and how it can be improved by using natural frequencies). Therefore the fact that, in the present study, people do not fully use this information, or use it inaccurately, may reflect the mode of information delivery.

In the response to this comment the authors have pointed out their own previous work showing that system neglect can occur even when numerical probabilities are not used. This is reassuring but there remains a large body of classic work showing that observers do struggle with conditional probabilities of the type presented in the task,

(2) Although a very precise model of 'system neglect' is presented, many other models could fit the data.

For example, you would get similar effects due to attraction of parameter estimates towards a global mean - essentially application of a hyper-prior in which the parameters applied by each participant in each block are attracted towards the experiment-wise mean values of these parameters. For example, the prior probability of regime shift ground-truth values [0.01, 0.05, 0.10] are mapped to subjective values of [0.037, 0.052, 0.069]; this would occur if observers apply a hyper-prior that the probability of regime shift is about 0.05 (the average value over all blocks). This 'attraction to the mean' is a well-established phenomenon and cannot be ruled out with the current data (I suppose you could rule it out by comparing to another dataset in which the mean ground-truth value was different).

More generally, any model in which participants don't fully use the numerical information they were given would produce apparent 'system neglect'. Four qualitatively different example reasons are: 1. Some individual participants completely ignored the probability values given. 2. Participants did not ignore the probability values given, but combined them with a hyperprior as above. 3. Participants had a reporting bias where their reported beliefs that a regime-change had occurred tend to be shifted towards 50% (rather than reporting 'confident' values such 5% or 95%). 4. Participants underweighted probability outliers resulting in underweighting of evidence in the 'high signal diagnosticity' environment (10.1016/j.neuron.2014.01.020)

In summary I agree that any model that fits the data would have to capture the idea that participants don't differentiate between the different environments as much as they should, but I think there are a number of qualitatively different reasons why they might do this - of which the above are only examples.

---

## [Author Response]

The following is the authors’ response to the previous reviews

**eLife Assessment**
This study offers valuable insights into how humans detect and adapt to regime shifts, highlighting dissociable contributions of the frontoparietal network and ventromedial prefrontal cortex to sensitivity to signal diagnosticity and transition probabilities. The combination of an innovative instructed-probability task, Bayesian behavioural modeling, and model-based fMRI analyses provides a solid foundation for the main claims; however, major interpretational limitations remain, particularly a potential confound between posterior switch probability and time in the neuroimaging results. At the behavioural level, reliance on explicitly instructed conditional probabilities leaves open alternative explanations that complicate attribution to a single computational mechanism, such that clearer disambiguation between competing accounts and stronger control of temporal and representational confounds would further strengthen the evidence.

Thank you. In this revision, we addressed Reviewer 3’s remaining concern on the potential confound between posterior probability and time in neuroimaging results. First, as suggested by the reviewer, we provided images of activations for the effect of Pt and delta Pt after controlling for intertemporal prior in GLM-2. Second, we compared the effect of Pt and delta Pt between GLM-1 (without intertemporal prior) and GLM-2 (with intertemporal prior) and showed the results in a new figure (Figure 4).

Regarding issue on reliance on explicitly instructed probabilities, we wish to point out that most of the concerns such as response mode and regression to the mean were addressed in the original behavioral paper by Massey and Wu (2005). Please see our response to this point in detail in Weakness (2) posted by Reviewer 3.

**Public Reviews:**

**Reviewer #1 (Public review):**
Summary:The study examines human biases in a regime-change task, in which participants have to report the probability of a regime change in the face of noisy data. The behavioral results indicate that humans display systematic biases, in particular, overreaction in stable but noisy environments and underreaction in volatile settings with more certain signals. fMRI results suggest that a frontoparietal brain network is selectively involved in representing subjective sensitivity to noise, while the vmPFC selectively represents sensitivity to the rate of change.Strengths:- The study relies on a task that measures regime-change detection primarily based on descriptive information about the noisiness and rate of change. This distinguishes the study from prior work using reversal-learning or change-point tasks in which participants are required to learn these parameters from experiences. The authors discuss these differences comprehensively.- The study uses a simple Bayes-optimal model combined with model fitting, which seems to describe the data well. The model is comprehensively validated.- The authors apply model-based fMRI analyses that provide a close link to behavioral results, offering an elegant way to examine individual biases.Weaknesses:The authors have adequately addressed my prior concerns.

Thank you for reviewing our paper and providing constructive comments that helped us improve our paper.

**Reviewer #3 (Public review):**

Thank you again for reviewing the manuscript. In this revision, we focused on addressing your concern on the potential confound between posterior probability and time in neuroimaging results. First, we presented whole-brain results of subjects’ probability estimates (Pt, their subjective posterior probability of switch) after controlling for the effect of time on probability of switch (the intertemporal prior). Second, we compared the effect of probability estimates (Pt) on vmPFC and ventral striatum activity—which we found to correlate with Pt—with and without including intertemporal prior in the GLM. These results will be summarized in a new figure (Figure 4) in the revised manuscript.

As suggested by the reviewer, we also added slice-by-slice images of the whole-brain results on Pt and delta Pt in the supplement in addition to the Tables of Activation so that the activated brain regions can be clearly seen through these images.

This study concerns how observers (human participants) detect changes in the statistics of their environment, termed regime shifts. To make this concrete, a series of 10 balls are drawn from an urn that contains mainly red or mainly blue balls. If there is a regime shift, the urn is changed over (from mainly red to mainly blue) at some point in the 10 trials. Participants report their belief that there has been a regime shift as a % probability. Their judgement should (mathematically) depend on the prior probability of a regime shift (which is set at one of three levels) and the strength of evidence (also one of three levels, operationalized as the proportion of red balls in the mostly-blue urn and vice versa). Participants are directly instructed of the prior probability of regime shift and proportion of red balls, which are presented on-screen as numerical probabilities. The task therefore differs from most previous work on this question in that probabilities are instructed rather than learned by observation, and beliefs are reported as numerical probabilities rather than being inferred from participants' choice behaviour (as in many bandit tasks, such as Behrens 2007 Nature Neurosci).The key behavioural finding is that participants over-estimate the prior probability of regime change when it is low, and under estimate it when it is high; and participants over-estimate the strength of evidence when it is low and under-estimate it when it is high. In other words participants make much less distinction between the different generative environments than an optimal observer would. This is termed 'system neglect'. A neuroeconomic-style mathematical model is presented and fit to data.Functional MRI results how that strength of evidence for a regime shift (roughly, the surprise associated with a blue ball from an apparently red urn) is associated with activity in the frontal-parietal orienting network. Meanwhile at time-points where the probability of a regime shift is high, there is activity in another network including vmPFC. Both networks show individual differences effects, such that people who were more sensitive to strength of evidence and prior probability show more activity in the frontal-parietal and vmPFC-linked networks respectively.Strengths(1) The study provides a different task for looking at change-detection and how this depends on estimates of environmental volatility and sensory evidence strength, in which participants are directly and precisely informed of the environmental volatility and sensory evidence strength rather than inferring them through observation as in most previous studies(2) Participants directly provide belief estimates as probabilities rather than experimenters inferring them from choice behaviour as in most previous studies(3) The results are consistent with well-established findings that surprising sensory events activate the frontal-parietal orienting network whilst updating of beliefs about the word ('regime shift') activates vmPFC.Weaknesses(1) The use of numerical probabilities (both to describe the environments to participants, and for participants to report their beliefs) may be problematic because people are notoriously bad at interpreting probabilities presented in this way, and show poor ability to reason with this information (see Kahneman's classic work on probabilistic reasoning, and how it can be improved by using natural frequencies). Therefore the fact that, in the present study, people do not fully use this information, or use it inaccurately, may reflect the mode of information delivery.In the response to this comment the authors have pointed out their own previous work showing that system neglect can occur even when numerical probabilities are not used. This is reassuring but there remains a large body of classic work showing that observers do struggle with conditional probabilities of the type presented in the task.

Thank you. Yes, people do struggle with conditional probabilities in many studies. However, as our previous work suggested (Massey and Wu, 2005), system-neglect was likely not due to response mode (having to enter probability estimates or making binary predictions, and etc.).

(2) Although a very precise model of 'system neglect' is presented, many other models could fit the data.For example, you would get similar effects due to attraction of parameter estimates towards a global mean - essentially application of a hyper-prior in which the parameters applied by each participant in each block are attracted towards the experiment-wise mean values of these parameters. For example, the prior probability of regime shift ground-truth values [0.01, 0.05, 0.10] are mapped to subjective values of [0.037, 0.052, 0.069]; this would occur if observers apply a hyper-prior that the probability of regime shift is about 0.05 (the average value over all blocks). This 'attraction to the mean' is a well-established phenomenon and cannot be ruled out with the current data (I suppose you could rule it out by comparing to another dataset in which the mean ground-truth value was different).More generally, any model in which participants don't fully use the numerical information they were given would produce apparent 'system neglect'. Four qualitatively different example reasons are: 1. Some individual participants completely ignored the probability values given. 2. Participants did not ignore the probability values given, but combined them with a hyperprior as above. 3. Participants had a reporting bias where their reported beliefs that a regime-change had occurred tend to be shifted towards 50% (rather than reporting 'confident' values such 5% or 95%). 4. Participants underweighted probability outliers, resulting in underweighting of evidence in the 'high signal diagnosticity' environment (10.1016/j.neuron.2014.01.020)In summary I agree that any model that fits the data would have to capture the idea that participants don't differentiate between the different environments as much as they should, but I think there are a number of qualitatively different reasons why they might do this - of which the above are only examples - hence I find it problematic that the authors present the behaviour as evidence for one extremely specific model.

We thank the reviewer for this comment. We thank you for putting out that there are alternative models that can describe the over- and underreaction seen in the dataset. Massey and Wu (2005) dealt with this possibility in their original paper. Their concern was not so much about alternative ways of modeling their results, but in terms of alternative psychological processes. For example, asymmetric noise accounts have been posited in the judgment and decision making literature as possible accounts of phenomena like over-confidence. They addressed what might be crudely called “regression/attraction to the mean” in two ways. First, they looked at median responses as well as mean responses (because medians are less affected by the regressive effect) and found the same patterns of over- and underreactions. Second, they also generated sequences that matched particular posterior probabilities (so that over- and underreaction cannot be explained by regression to the mean) and still found under- and overreactions.

We also wish to point out in the judgment and decision making literature starting from Edwards (1968), there is a long history of using normative Bayesian model as the starting model and subsequently develop quasi-Bayesian models (like the system-neglect model) to describe systematic deviations from the normative Bayesian.

Finally, we want to clarify that our primary goal is not to engage in model fitting exercise that examines different possible models. To us, what is more important is that system neglect is a psychologically motivated hypothesis. It is built on the idea that the lack of sensitivity to the system parameters is due to the fact that people focus primarily on the signals and secondarily on the system parameters that generate the signals. Massey and Wu (2005) dealt with a host of other potential explanations through experimental manipulations and data analysis. In this paper, we built on Massey and Wu to examine the neurocomputational basis that gives rise to over- and underreactions.

(3) Despite efforts to control confounds in the fMRI study, including two control experiments, I think some confounds remain.For example, a network of regions is presented as correlating with the cumulative probability that there has been a regime shift in this block of 10 samples (Pt). However, regardless of the exact samples shown, Pt always increases with sample number (as by the time of later samples, there have been more opportunities for a regime shift)? To control for this the authors include, in a supplementary analysis, an 'intertemporal prior.' I would have preferred to see the results of this better-controlled analysis presented in the main figure. From the tables in the SI it is very difficult to tell how the results change with the includion of the control regressors.

Thank you. In response, we added a new figure, now Figure 4, showing the results of Pt and delta Pt from GLM-2 where we added the intertemporal prior as a regressor to control for temporal confounds. We compared Pt and delta Pt results in vmPFC and ventral striatum between GLM-1 and GLM-2. We also showed the results on intertemporal prior on vmPFC and ventral striatum from GLM-2.

On the other hand, two additional fMRI experiments are done as control experiments and the effect of Pt in the main study is compared to Pt in these control experiments. Whilst I admire the effort in carrying out control studies, I can't understand how these particular experiment are useful controls. For example, in experiment 3 participants simply type in numbers presented on the screen - how can we even have an estimate of Pt from this task?

We thank the reviewer for this comment. On the one hand, the effect of Pt we see in brain activity can be simply due to motor confounds and the purpose of Experiment 3 was to control for them. Our question was, if subjects saw the similar visual layout and were just instructed to press buttons to indicate two-digit numbers, would we observe the vmPFC, ventral striatum, and the frontoparietal network like what we did in the main experiment (Experiment 1)?

On the other hand, the effect of Pt can simply reflect probability estimates of that the current regime is the blue regime, and therefore not particularly about change detection. In Experiment 2, we tested that idea, namely whether what we found about Pt was unique to change detection. In Experiment 2, subjects estimated the probability that the current regime is the blue regime (just as they did in Experiment 1) except that there were no regime shifts involved. In other words, it is possible that the regions we identified were generally associated with probability estimation and not particularly about probability estimates of change. We used Experiment 2 to examine whether this were true.

To make the purpose of the two control experiments clearer, we updated the paragraph describing the control experiments on page 9:

“To establish the neural representations for regime-shift estimation, we performed three fMRI experiments (*n* = 30 subjects for each experiment, 90 subjects in total). Experiment 1 was the main experiment, while Experiments 2 to 3 were control experiments that ruled out two important confounds (Fig. 1E). The control experiments were designed to clarify whether any effect of subjects’ probability estimates of a regime shift, *Pt*, in brain activity can be uniquely attributed to change detection. Here we considered two major confounds that can contribute to the effect of *Pt*. First, since subjects in Experiment 1 made judgments about the probability that the current regime is the blue regime (which corresponded to probability of regime change), the effect of *Pt* did not particularly have to do with change detection. To address this issue, in Experiment 2 subjects made exactly the same judgments as in Experiment 1 except that the environments were stationary (no transition from one regime to another was possible), as in Edwards (1968) classic “bookbag-and-poker chip” studies. Subjects in both experiments had to estimate the probability that the current regime is the blue regime, but this estimation corresponded to the estimates of regime change only in Experiment 1. Therefore, activity that correlated with probability estimates in Experiment 1 but not in Experiment 2 can be uniquely attributed to representing regime-shift judgments. Second, the effect of *Pt* can be due to motor preparation and/or execution, as subjects in Experiment 1 entered two-digit numbers with button presses to indicate their probability estimates. To address this issue, in Experiment 3 subjects performed a task where they were presented with two-digit numbers and were instructed to enter the numbers with button presses. By comparing the fMRI results of these experiments, we were therefore able to establish the neural representations that can be uniquely attributed to the probability estimates of regime-shift.”

To further make sure that the probability-estimate signals in Experiment 1 were not due to motor confounds, we implemented an action-handedness regressor in the GLM, as we described below on page 19:

“Finally, we note that in GLM-1, we implemented an “action-handedness” regressor to directly address the motor-confound issue, that higher probability estimates preferentially involved right-handed responses for entering higher digits. The action-handedness regressor was parametric, coding -1 if both finger presses involved the left hand (e.g., a subject pressed “23” as her probability estimate when seeing a signal), 0 if using one left finger and one right finger (e.g., “75”), and 1 if both finger presses involved the right hand (e.g., “90”). Taken together, these results ruled out motor confounds and suggested that vmPFC and ventral striatum represent subjects’ probability estimates of change (regime shifts) and belief revision.”

(4) The Discussion is very long, and whilst a lot of related literature is cited, I found it hard to pin down within the discussion, what the key contributions of this study are. In my opinion it would be better to have a short but incisive discussion highlighting the advances in understanding that arise from the current study, rather than reviewing the field so broadly.

Thank you. We thank the reviewer for pushing us to highlight the key contributions. In response, we added a paragraph at the beginning of Discussion to better highlight our contributions:

“In this study, we investigated how humans detect changes in the environments and the neural mechanisms that contribute to how we might under- and overreact in our judgments. Combining a novel behavioral paradigm with computational modeling and fMRI, we discovered that sensitivity to environmental parameters that directly impact change detection is a key mechanism for under- and overreactions. This mechanism is implemented by distinct brain networks in the frontal and parietal cortices and in accordance with the computational roles they played in change detection. By introducing the framework in system neglect and providing evidence for its neural implementations, this study offered both theoretical and empirical insights into how systematic judgment biases arise in dynamic environments.”

**Recommendations for the authors:**

**Reviewer #3 (Recommendations for the authors):**
Thank you for pointing out the inclusion of the intertemporal prior in glm2, this seems like an important control that would address my criticism. Why not present this better-controlled analysis in the main figure, rather than the results for glm1 which has no effective control of the increasing posterior probability of a reversal with time?

Thank you for this suggestion. We added a new figure (Figure 4) that showed results of Pt and delta Pt from GLM-2. We also compared the effect of Pt and delta Pt between GLM-1 and GLM-2. We found that the effect of Pt and delta Pt did not differ between GLM-1 and GLM-2. GLM-1 and GLM-2 differed on whether various task-related regressors contributing to Pt, including the intertemporal prior, were included in the model. In GLM-1, those task-related regressors were not included. In GLM-2, the task-related regressors were included in addition to Pt and delta P.

The reason we kept results from GLM-1 (Figure 3) was primarily because we wanted to compare the effect of Pt between experiments under identical GLM. In other words, the regressors in GLM-1 was identical across all 3 experiments. In Experiments 1 and 2, Pt and delta Pt were respectively probability estimates and belief updates that current regime was the Blue regime. In Experiment 3, Pt and delta Pt were simply the number subjects were instructed to press (Pt) and change in number between successive periods (delta Pt).

Here is the section in the main text where we discussed the new Figure 4 on page 19-22:

We further examined the robustness of *Pt* and ∆*Pt* representations in vmPFC and ventral striatum in three follow-up analyses. In the first analysis, we implemented a GLM (GLM-2 in Methods) that, in addition to *Pt* and ∆*Pt*, included various task-related variables contributing to *Pt* as regressors. Specifically, to account for the fact that the probability of regime change increased over time, we included the intertemporal prior as a regressor in GLM-2. The intertemporal prior is the natural logarithm of the odds in favor of regime shift in the *t*-th period, \begin{document}$\ln \left(\frac{1-(1-q)^{t}}{(1-q)^{t}}\right)$\end{document}, where *q* is transition probability and *t* = 1, …, 10is the period (Eq. 1 in Methods). It describes normatively how the prior probability of change increased over time regardless of the signals (blue and red balls) the subjects saw during a trial. Including it along with *Pt* would clarify whether any effect of *Pt* can otherwise be attributed to the intertemporal prior. We found that the results of *Pt* and ∆*Pt* in the vmPFC and ventral striatum in GLM-2 were identical to those in GLM-1 (Fig. 4): Fig. 4A was meant to depict the results in slices identical to those shown in Fig. 3B for results based on GLM-1. For slice-by-slice results, see Fig. S7 in SI for results based on GLM-1 and Fig. S9 for GLM-2. For Tables of activations, see Tables S1-S3 in *SI* for GLM-1 and Tables S7-S9 for GLM-2. In a separate, independent region-of-interest (ROI) analysis on vmPFC and ventral striatum (Fig. 4BC; see Independent regions-of-interest (ROIs) analysis in Methods for details), we further compared the effect of both *Pt* and ∆*Pt* between GLM-1 and GLM-2. For *Pt*, the difference between GLM-1 and GLM-2 was not significant (paired t-test, *t*(58) = −0.72, *p* = 0.47 in vmPFC, *t*(58) = −0.21, *p* = 0.83 in ventral striatum), while the effect of *Pt* from GLM-1 (one sample t-test, *t*(29) = −3,82, *p* <.01 in vmPFC; *t*(29) = −3.06, *p* <.01 in ventral striatum) and GLM-2 was significant (one-sample t-test, *t*(29) = −2.69, *p* = .01 in vmPFC; *t*(29) = −2.50, *p* .02 in ventral striatum). For ∆*Pt*, the difference between GLM-1 and GLM-2 was not significant (paired t-test, *t*(58) = −0.07, *p* = 0.94 in vmPFC; *t*(58) = −0.14, *p* = 0.88 in ventral striatum), while the effect of from GLM-1 (one-sample t-test, *t*(29) = −3.12, *p* <.01 in vmPFC; *t*(29) = −4.14, *p* <.01 in ventral striatum) and GLM-2 was significant (one-sample t-test, *t*(29) = −2.92, *p* <.01 in vmPFC; *t*(29) = −3.59, *p* <.01 in ventral striatum). For the intertemporal prior, activity in both vmPFC and ventral striatum did not correlate significantly with the intertemporal prior (one-sample t-test, *t*(29) = −0.07, *p* = 0.95 in vmPFC; *t*(29) = −0.53, *p* = 0.60 in ventral striatum). All the t-tests described above were two-tailed. Taken together, these results suggest that vmPFC and ventral striatum represented *Pt* and ∆*Pt* regardless of whether the intertemporal prior and other task-related regressors contributing to *Pt* were included in the GLM. We also did not find that vmPFC and ventral striatum to represent the intertemporal prior. In the second analysis, we implemented a GLM that replaced *Pt* with the log odds of *Pt*, 1n (*Pt*/(1 - *Pt*)) (Fig. S10 in *SI*). In the third analysis, we implemented a GLM that examined *Pt* separately on periods when change-consistent (blue balls) and change-inconsistent (red balls) signals appeared (Fig. S11 in SI). Each of these analyses showed significant correlation with *Pt* in vmPFC and ventral striatum, further establishing the robustness of the *Pt* findings.

As a further point I could not navigate the tables of fMRI activations in SI and recommend replacing or supplementing these with images. For example I cannot actually find a vmPFC or ventral striatum cluster listed for the effect of Pt in GLM1 (version in table S1), which I thought were the main results? Beyond that, comparing how much weaker (or not) those results are when additional confound regressors are included in GLM2 seems impossible.

As suggested by the reviewer, we added slice-by-slice images showing the effect of Pt and delta Pt (Figure S9 in SI for GLM-2 and Figure S7 for GLM-1). The clusters in blue represent Pt effect, the clusters in orange represent delta Pt effect. As can be seen, both Pt and delta Pt are represented in the vmPFC and ventral striatum.